# Structured Voronoi Sampling

**Afra Amini**[1]     **Li Du**[2]     **Ryan Cotterell**[1]
[1]ETH Zürich     [2]Johns Hopkins University
{afra.amini, ryan.cotterell}@inf.ethz.ch
leodu@cs.jhu.edu

## Abstract

Gradient-based sampling algorithms have demonstrated their effectiveness in text generation, especially in the context of controlled text generation. However, there exists a lack of theoretically grounded and principled approaches for this task. In this paper, we take an important step toward building a principled approach for sampling from language models with gradient-based methods. We use discrete distributions given by language models to define densities and develop an algorithm based on Hamiltonian Monte Carlo to sample from them. We name our gradient-based technique Structured Voronoi Sampling (SVS). In an experimental setup where the reference distribution is known, we show that the empirical distribution of SVS samples is closer to the reference distribution compared to alternative sampling schemes. Furthermore, in a controlled generation task, SVS is able to generate fluent and diverse samples while following the control targets significantly better than other methods.

 https://github.com/AfraAmini/svs

## 1 Introduction

Gradient-based sampling algorithms such as Hamiltonian Monte Carlo [HMC; 32] and Langevin dynamics [46] are widely used in Bayesian inference due to their efficiency in drawing samples from high-dimensional space [4]. Such algorithms construct a Markov Chain that has the desired distribution as its stationary distribution and use the gradient information of this distribution to efficiently navigate the state space. Additionally, gradient-based sampling schemes have recently been deployed in computer vision to generate high-quality images from state-of-the-art models [13, 43], and are a popular choice for tasks such as image synthesis [5] and image-to-image translation [40].

In natural language processing, there have been several attempts to apply gradient-based sampling techniques to sampling text from neural language models [21, 23, 38]. The motivation behind this approach to text generation is to sample from energy-based probabilistic models, where the normalization factor is not tractable. One such case is in controlled text generation, where energy functions are usually defined as a linear combination of LM probabilities and the probability of satisfying a set of predefined constraints [21, 23, 38]. In contrast to computer vision, however, applying gradient-based sampling schemes to text generation is nuanced as text, in contrast to images, is discrete.

Upon closer inspection, none of the proposed algorithms actually defines a valid Markov Chain Monte Carlo [MCMC; 12, 29] scheme that will draw samples from the model in the limit. For instance, Qin et al. [38] relax the language model from a distribution over strings to a distribution over logits. While the relaxation does transform the language model into a continuous distribution, it introduces bias. Kumar et al. [MUCOLA; 21] take a different approach. They derive a constrained gradient-based

sampler where the constraint is enforced through a projection. However, the projection invalidates the MCMC procedures, leading to an algorithm without guarantees.[1]

We derive a principled Hamiltonian Monte Carlo scheme for generating text from language models, which we call a structured Voronoi sampler. Our scheme consists of two steps. First, we give a recipe for encoding discrete distributions as densities over $\mathbb{R}^d$; we term the resulting encoding Voronoi measures.[2] Second, we derive a refractive Hamiltonian Monte Carlo algorithm [30] for sampling from an arbitrary Voronoi measure. In our theoretical analysis, we show that, despite the presence of discontinuities, we are able to give proof that our sampler satisfies the detailed balance condition and, thus, is a correct MCMC scheme.

To empirically evaluate the performance of structured Voronoi sampling, we begin by applying it to a toy example where the exact reference probability distribution is known. We compare the empirical distribution of drawn samples with the reference distribution and show Voronoi sampler's distribution is closer to the reference distribution than MUCOLA or unconstrained HMC. Furthermore, we use our sampling scheme for controlled generation, where the goal is to use GPT-2 to generate restaurant reviews for a target type of food, e.g., Italian, Fast food, or Japanese, and separately to generate text with a positive sentiment. We find that structured Voronoi sampling outperforms FUDGE [48], MUCOLA, and Langevin Dynamics algorithms in terms of adhering to the control target. Additionally, the samples generated by structured Voronoi sampling are comparably fluent and diverse to those produced by the other methods.

## 2 Language Models

Let $\Sigma$ be an alphabet, a finite, non-empty set. By $\Sigma^* \overset{\text{def}}{=} \bigcup_{n=0}^{\infty} \Sigma^n$, we denote the Kleene closure of $\Sigma$.[3] A probability distribution over $\Sigma^*$ is called a **language model** (**LM**). The elements of $\Sigma$ may be characters, subword pieces, or words; the choice lies with the modeler. Language models can be factored autoregressively by means of the chain rule of probability, i.e., for any string $\boldsymbol{w} = w_1 \cdots w_N \in \Sigma^*$, we can write

$$p(\boldsymbol{w}) = p(\text{EOS} \mid \boldsymbol{w}) \prod_{n=1}^{N} p(w_n \mid \boldsymbol{w}_{<n}), \tag{1}$$

where $\text{EOS} \notin \Sigma$ is a distinguished end-of-sequence token and $\boldsymbol{w}_{<n}$ is the prefix of length $(n-1)$ of the string $\boldsymbol{w}$. We define $\overline{\Sigma} \overset{\text{def}}{=} \Sigma \cup \{\text{EOS}\}$, and require the conditional distributions $p(\cdot \mid \boldsymbol{w}_{<n})$ to be defined over $\overline{\Sigma}$. While all language models can be factored autoregressively, not all conditionals $p(\cdot \mid \boldsymbol{w}_{<n})$ can be assembled into a language model. In some cases, probability mass must be placed on infinite sequences [6]. In this work, we assume working with **tight** language models, i.e., that they indeed define valid probability distributions over $\Sigma^*$.

### 2.1 Language Modeling with Embeddings

Most neural language models make use of embeddings. Moreover, in most language models, the weights are shared between the language model head and the embedding layer. Such an embedding-based language model is defined as follows

$$p(w_n \mid \boldsymbol{w}_{<n}) \overset{\text{def}}{=} \frac{\exp \mathbf{v}_{w_n} \cdot \text{enc}(\boldsymbol{w}_{<n})}{\sum_{w \in \overline{\Sigma}} \exp \mathbf{v}_w \cdot \text{enc}(\boldsymbol{w}_{<n})}, \tag{2}$$

where $\mathbf{v}_w \in \mathbb{R}^d$ is the embedding of $w$ and $\text{enc} : \Sigma^* \to \mathbb{R}^d$ is a real-valued encoding of the context. Notably, the context embedding $\text{enc}(\boldsymbol{w}_{<n}) \in \mathbb{R}^d$ is obtained by inputting the context $\boldsymbol{w}_{<n}$ into the language model, converting it to embeddings $\mathbf{v}_{\boldsymbol{w}_{<n}}$, passing it through neural network layers, and extracting the encoding from the model at position $n-1$.

---

[1]In fact, a similar projection step is often applied in computer vision applications, which motivated Lou and Ermon [26] to develop a principled approach that avoids projection.

[2]We call the measures structured Voronoi samplers when the state face is larger and factored.

[3]We define $\Sigma^0 \overset{\text{def}}{=} \{\varepsilon\}$.

Interestingly, one can lift an embedding-based language model to be a distribution over the set of **base embeddings**: $\mathcal{B} = \{\mathbf{v}_w : w \in \overline{\Sigma}\}$. Specifically, we can associate a sequence of embeddings $\mathbf{V} = [\mathbf{v}_1, \ldots, \mathbf{v}_N]$ to any string $\boldsymbol{w} = w_1 \cdots w_N$. Substituting each $w$ with the corresponding $\mathbf{v}_w$, we can rewrite Eq. (1) as

$$p(\mathbf{V}) = p(\mathbf{v}_{\text{EOS}} \mid \mathbf{V}) \prod_{n=1}^{N} p(\mathbf{v}_n \mid \mathbf{V}_{<n}). \tag{3}$$

This transformation encodes a language model as a distribution over real-valued vectors. However, $p(\mathbf{V})$ only places a positive probability on a countable set, and is zero everywhere else.

## 2.2 Controlled Language Modeling with Embeddings

In a controlled generation task, we are interested in a subset of strings $\boldsymbol{w}$ that have a target property $t$. Therefore, we want to model and sample from a conditional distribution $p(\boldsymbol{w} \mid t)$, e.g., sample a sentence given a topic. Following Bayes' rule, one can write $p(\boldsymbol{w} \mid t) \propto p(t \mid \boldsymbol{w}) \, p(\boldsymbol{w})$. Previous papers model $p(t \mid \boldsymbol{w})$ with an embedding-augmented classifier. Such a classifier receives embeddings $\mathbf{V}$ associated with $\boldsymbol{w}$ and predicts the probability of the target $t$. Notably, if the classifier and the LM share the same base embeddings $\mathcal{B}$, the controlled LM can also be lifted as a distribution over the base embeddings

$$p(\mathbf{V} \mid t) = \frac{1}{Z_t} \, p(t \mid \mathbf{V}) \, p(\mathbf{V}), \tag{4}$$

where $Z_t = \sum_{\mathbf{V}} p(t \mid \mathbf{V}) \, p(\mathbf{V})$ is an intractable normalization factor, that sums over the embeddings of all possible strings.[4] Identically to $p(\mathbf{V})$, $p(\mathbf{V} \mid t)$ only places a positive probability on a countable set.

## 3 Voronoi Measures

In this section, we demonstrate how to encode an embedding-based language model as a density that places positive probability on a set with a measure greater than zero. Such encoding allows us to derive a principled gradient-based sampling approach to generate samples in §6. We start with some definitions.

**Definition 1.** *An **embedding-augmented** probability distribution over the first $M$ positive integers $[M]$ is an array $\boldsymbol{p} = [p_1, \ldots, p_M]$ such that $p_m \geq 0$ and $\sum_{m=1}^{M} p_m = 1$ where we assume that there is a real-valued embedding $\{\mathbf{v}_m\}_{m=1}^{M} \subset \mathbb{R}^d$ associated with each $m \in [M]$.*

Embedding-augmented distributions can be viewed as densities over $\mathbb{R}^d$ using the following simple encoding

$$p(\mathbf{x}) = \begin{cases} p_m, & \text{if } \mathbf{x} = \mathbf{v}_m \\ 0, & \text{otherwise}. \end{cases} \tag{5}$$

Eq. (5), however, yields a density that is $0$ almost everywhere (with respect to the standard Lebesgue measure) and its gradient with respect to $p_m$ is also zero almost everywhere. Thus, Eq. (5) is not amenable to gradient-based sampling, and to derive a meaningful gradient-based sampling we require a more nuanced encoding.

To provide such an encoding, we introduce the **Voronoi measure**. Given an embedding-augmented distribution $\boldsymbol{p} = [p_1, \ldots, p_M]$ with embeddings $\{\mathbf{v}_m\}_{m=1}^{M}$, and a compact set $\mathcal{K} \subset \mathbb{R}^d$ that covers the embeddings, i.e., $\{\mathbf{v}_m\}_{m=1}^{M} \subset \mathcal{K}$, we define the **Voronoi cell** for the $m^{\text{th}}$ item with respect to the compact set $\mathcal{K}$ as follows

$$C_m = \left\{ \mathbf{x} : \mathbf{x} \in \mathcal{K}, ||\mathbf{x} - \mathbf{v}_m||_2^2 \leq ||\mathbf{x} - \mathbf{v}_{m'}||_2^2, \forall m' \neq m \right\}. \tag{6}$$

Now, using the definition of a Voronoi cell $C_m$ given in Eq. (6), we can define a density that is *not* zero almost everywhere as follows. The strategy is to spread out the probability mass $p_m$ over the entirety of the set $C_m$. To do so, we assume access to a set of **base measures** $\{\mu_m\}_{m=1}^{M}$ that give us a reference for how to judge the probability mass in each $C_m$. We make this encoding formal in the following definition.

---

[4]We will discuss in §5.2 how gradient-based sampling can help to sample from this distribution without the need to compute $Z_t$.

**Definition 2.** *Let $\boldsymbol{p} = [p_1, \ldots, p_M]$ be an embedding-augmented distribution with embeddings $\{\mathbf{v}_m\}_{m=1}^M \subset \mathbb{R}^d$, and let $\mathcal{K}$ be a compact set such that $\{\mathbf{v}_m\}_{m=1}^M \subset \mathcal{K}$. Furthermore, let $\{\mu_m\}_{m=1}^M$ be a set of base measures over $\mathbb{R}^d$ that are absolutely continuous with respect to the standard Lebesgue measure $\lambda$ over $\mathbb{R}^d$, i.e., $\mu_m \ll \lambda$. Define the $(\mathcal{K}, \mu)$-**Voronoi measure** as follows*

$$p_{\mathrm{V}}(\mathbf{x}) \stackrel{\text{def}}{=} \begin{cases} \frac{p_{m^\star(\mathbf{x})}}{\mu_m(C_{m^\star(\mathbf{x})})} \frac{\mathrm{d}\mu_m}{\mathrm{d}\lambda}(\mathbf{x}), & \textbf{if } \mathbf{x} \in \mathcal{K} \\ 0, & \textbf{otherwise} \end{cases} \tag{7}$$

*where we define projection*

$$m^\star(\mathbf{x}) \stackrel{\text{def}}{=} \operatorname*{argmin}_{m \in [M]} ||\mathbf{x} - \mathbf{v}_m||_2^2. \tag{8}$$

*with ties broken arbitrarily.*[5]

In the following proposition, we make precise the sense in which a Voronoi measure encodes the original embedding-augmented distribution.

**Proposition 1.** *Let $\boldsymbol{p} = [p_1, \ldots, p_M]$ be an embedding-augmented distribution with embeddings $\{\mathbf{v}_m\}_{m=1}^M \subset \mathbb{R}^d$, and let $p_{\mathrm{V}}$ be the corresponding Voronoi measure Eq. (7). Then, $p_{\mathrm{V}}(C_m) = p_m$ where $C_m$ is defined as in Eq. (6). See App. C.1 for proof.*

**Example 1.** *Suppose $\boldsymbol{p} = [p_1, \ldots, p_4]$ is a categorical distribution, and there are 4 embeddings in $\mathbb{R}^2$ associated with each $p_i$, namely: $\mathbf{v}_1 = [1, 1], \mathbf{v}_2 = [-1, 1], \mathbf{v}_3 = [-1, -1], \mathbf{v}_4 = [1, -1]$. Given the $\mathcal{K} = [-2, 2] \times [-2, 2]$ and the embedding-augmented probability distribution $\boldsymbol{p}$, Eq. (7) defines a Voronoi measure over this space, where the Voronoi cells are visualized in Fig. 1. We will discuss in §6 how Voronoi sampling navigates this space.*

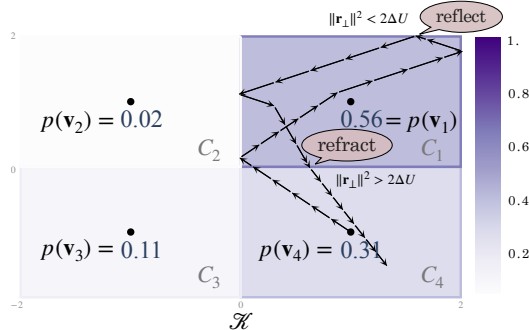

Figure 1: The example shows how Voronoi Sampling navigates through the space to sample one embedding in $\mathbb{R}^2$. Each Voronoi is annotated with the probability of its center, i.e., Voronoi measure of the cell.

### 3.1 Structured Voronoi Measures

To encode language models as densities more naturally, we introduce a generalization of the Voronoi measure, which we term a structured Voronoi measure. Now, rather than a distribution over $M$ elements, we assume to have a sequence of length $N$. Each token in the sequence takes value in $[M]$. Let $\boldsymbol{m} = [m_1, \ldots, m_N] \in [M]^N$. We define a **structured Voronoi cell** as $C_{\boldsymbol{m}} = \prod_{n=1}^N C_{m_n}$, where $\prod$ denotes the Cartesian product and we define the individual Voronoi cell as

$$C_{m_n} = \left\{ \mathbf{x} : \mathbf{x} \in \mathcal{K}, ||\mathbf{x} - \mathbf{v}_{m_n}||_2^2 \leq ||\mathbf{x} - \mathbf{v}_{m'}||_2^2, \forall m' \neq m_n \right\}. \tag{9}$$

**Proposition 2.** *Let $\mu$ be a measure on $\mathbb{R}^d$. Then, we have the product measure space as $\mu(C_{\boldsymbol{m}}) = \prod_{n=1}^N \mu(C_{m_n})$. See App. C.2 for proof.*

**Definition 3.** *Let $\boldsymbol{p}$ be an embedding-augmented distribution over $[M]^N$. For $\boldsymbol{m} \in [M]^N$, we denote $\boldsymbol{m}$'s probability as $p_{\boldsymbol{m}}$, and $\boldsymbol{m}$'s embedding as $\mathbf{V}_{\boldsymbol{m}} \in \mathbb{R}^{N \times d}$. Let $\mathcal{K}$ be a compact set that covers the embeddings $\mathbf{V}_{\boldsymbol{m}}$ and let $\mu \ll \lambda$ be a base measure absolutely continuous with respect to the Lebesgue measure $\lambda$. We define the $(\mathcal{K}, \mu)$-**structured Voronoi measure** as follows*

$$p_{\mathrm{V}}(\mathbf{x}) \stackrel{\text{def}}{=} \begin{cases} \frac{p_{\boldsymbol{m}^\star(\mathbf{x})}}{\mu(C_{\boldsymbol{m}^\star(\mathbf{x})})} \frac{\mathrm{d}\mu}{\mathrm{d}\lambda}(\mathbf{x}), & \textbf{if } \mathbf{x} \in \mathcal{K} \\ 0, & \textbf{otherwise} \end{cases} \tag{10}$$

*where we define the structured projection*

$$\boldsymbol{m}^\star(\mathbf{x}) \stackrel{\text{def}}{=} \operatorname*{argmin}_{\boldsymbol{m} \in [M]^N} \sum_{n=1}^N ||\mathbf{x}_n - \mathbf{V}_{\boldsymbol{m}_n}||_2^2. \tag{11}$$

---

[5]The set of $\mathbf{x}$ that induces a tie is a set of measure zero under the Lebesgue measure.

# 4 Application to Text Generation

Def. 3 gives us the flexibility to use any probability distribution over sequences of embeddings to define a structured Voronoi measure. For example, one can substitute $p_{m^\star(\mathbf{x})}$ with an embedding-augmented LM, i.e., Eq. (3), to encode a language model as a structured Voronoi measure. Another example is to encode a controlled LM as a structured Voronoi measure by substituting $p_{m^\star(\mathbf{x})}$ with Eq. (4).

**Base Measure.** We have explained how to encode our desired distribution as a structured Voronoi measure. However, in order to actually implement a gradient-based sampler, we need to specify the base probability measures $\{\mu_m\}_{m=1}^M$. Given an embedding-augmented probability $p(\mathbf{V})$ it is natural to follow the gradient of $\log p(\mathbf{V})$ with respect to the word embedding, i.e., $\mathbf{g}_m = \nabla_{\mathbf{V}} \log p(\mathbf{V})$. Thus, if we want to follow a direction similar to $\mathbf{g}_m$, one natural choice for $\mu$ is a Gaussian measure centered at the gradient $\mathbf{g}_m$ *restricted* to Voronoi cell $C_m$,[6] which we define:

$$\mu(A) = \frac{1}{\mu(C_m)} \int_{A \cap C_m} \exp\left(-\frac{1}{2} \|\mathbf{g}_m - \mathbf{x}\|_2^2\right) d\lambda(\mathbf{x}). \tag{12}$$

The normalizer ensures the measure is a probability measure. Furthermore, we have that $\mu$'s Radon–Nikodym derivative with respect to the Lebesgue measure $\lambda$ is given by

$$\frac{d\mu}{d\lambda}(\mathbf{x}) \overset{\text{def}}{=} \begin{cases} \frac{1}{\mu(C_m)} \exp\left(-\frac{1}{2} \|\mathbf{g}_m - \mathbf{x}\|_2^2\right), & \textbf{if } \mathbf{x} \in C_m \\ 0, & \textbf{otherwise} \end{cases} \tag{13}$$

Eq. (13) should be recognizable as the standard Gaussian density, albeit one that is truncated to the Voronoi cell $C_m$ [27].

**Proposition 3.** *Eq.* (13) *is absolutely continuous with respect to the Lebesgue measure* $\lambda$. *See App. C.3 for proof.*

**Proposition 4.** *The gradient of the log of the Voronoi measure* $p_V$ *is given by*

$$\nabla_{\mathbf{x}} \log p_V(\mathbf{x}) = \begin{cases} \mathbf{g}_m - \mathbf{x}, & \textbf{if } \mathbf{x} \in \text{int}(C_m) \\ undefined, & \textbf{if } \mathbf{x} \in \partial C_m \\ \mathbf{0}, & \textbf{otherwise} \end{cases} \tag{14}$$

*where the first two blocks in the case statement apply if there exists some* $m$ *such that* $\mathbf{x} \in \text{int}(C_m)$ *or* $\mathbf{x} \in \partial C_m$. *See App. C.4 for proof.*

# 5 Gradient-Based Sampling

In this section, we first review gradient-based sampling methods, namely HMC and Langevin dynamics. Then we discuss how they have been used in previous papers on text generation. This will set the stage for our algorithm in §6, which is based on HMC.

## 5.1 Hamiltonian Monte Carlo

The goal of HMC, originally proposed by Duane et al. [7], is to design a better proposal distribution in the standard Metropolis–Hastings MCMC by taking advantage of the gradient information in a principled way. Concretely, to sample from a given distribution $p(\mathbf{x})$ where $\mathbf{x} \in \mathbb{R}^d$, HMC treats $\mathbf{x}$ as the coordinates of the particles in some fictitious physical system. It then introduces an auxiliary momentum variable $\mathbf{r} \in \mathbb{R}^d$ associated with each coordinate and defines a Hamiltonian function $H(\mathbf{x}, \mathbf{r})$. Here, the Hamiltonian $H$ has the intuitive physical interpretation of the total energy of some conservative system, and, in classical mechanics, decomposes into the potential energy $U(\mathbf{x})$ and the kinetic energy $K(\mathbf{r})$, i.e., $H(\mathbf{x}, \mathbf{r}) = U(\mathbf{x}) + K(\mathbf{r})$. This formulation is convenient in part because if we define the joint distribution $p(\mathbf{x}, \mathbf{r}) \propto e^{-H(\mathbf{x}, \mathbf{r})}$ as in energy-based models, then

$$p(\mathbf{x}, \mathbf{r}) \propto e^{-U(\mathbf{x})} \cdot e^{-K(\mathbf{r})}, \tag{15}$$

---

[6]Please refer to App. B for a discussion on the reasoning behind choosing the Gaussian measure.

which means we can treat $\mathbf{x}$ and $\mathbf{r}$ as independent variables. Naturally, we can let $U(\mathbf{x}) = -\log p(\mathbf{x})$ so that $\mathbf{x}$ has the marginal of the target distribution. It is also common practice to set $K(\mathbf{r}) = \mathbf{r}^\top M^{-1}\mathbf{r}/2$ so that the momentum variable has a Gaussian distribution. Here, $M \in \mathbb{R}^{d \times d}$ is called the **mass matrix** and commonly set to identity.[7]

The Hamiltonian $H$ determines the equations of motion in a physical system, given by the Hamiltonian equations, which is also known as Hamiltonian dynamics,

$$\frac{\mathrm{d}\mathbf{x}}{\mathrm{d}t} = \frac{\partial H}{\partial \mathbf{r}}, \qquad \frac{\mathrm{d}\mathbf{r}}{\mathrm{d}t} = -\frac{\partial H}{\partial \mathbf{x}}. \qquad (16)$$

We are now ready to give a high-level description of how HMC generates a single sample (see Algorithm 1): First sample a momentum $\mathbf{r}$ from a Gaussian (line 1), then evolve the system using the Hamiltonian equations Eq. (16) for some predetermined amount of time (lines 4 to 7), and finally accept the new state with the Metropolis–Hastings acceptance criterion (lines 8 to 13). Note that the Hamiltonian equations often admit no closed-form solution in practice, and hence one needs to use numerical integrators for approximation. In particular, the leapfrog integrator, corresponding to lines 5 to 7, is almost always used in HMC.

---

**Algorithm 1** HMC

**Input:** $\mathbf{x}^t$: current sample, $U$: potential energy function, $\varepsilon$: step size, $L$: number of leapfrog steps
**Output:** next sample: $\mathbf{x}^{t+1}$

---

1: $\mathbf{r} \sim \mathcal{N}(\mathbf{0}, \mathbb{I})$
2: $H^t \leftarrow U(\mathbf{x}^t) + K(\mathbf{r})$
3: $\mathbf{x}^{t+1} \leftarrow \mathbf{x}^t$
4: **for** $l = 1, \dots, L$ :
5: $\quad \mathbf{r} \leftarrow \mathbf{r} - \frac{\varepsilon}{2}\nabla U(\mathbf{x}^{t+1})$
6: $\quad \mathbf{x}^{t+1} \leftarrow \mathbf{x}^{t+1} + \varepsilon\mathbf{r}$
7: $\quad \mathbf{r} \leftarrow \mathbf{r} - \frac{\varepsilon}{2}\nabla U(\mathbf{x}^{t+1})$
8: $H^{t+1} \leftarrow U(\mathbf{x}^{t+1}) + K(\mathbf{r})$
9: $\Delta H \leftarrow H^{t+1} - H^t$
10: **if** $\mathbf{s} \sim \mathcal{U}(0, 1) < e^{-\Delta H}$ :
11: $\quad$ **return** $\mathbf{x}^{t+1}$
12: **else**
13: $\quad$ **return** $\mathbf{x}^{t+1} \leftarrow \mathbf{x}^t$

---

Ingeniously designed by Duane et al. [7], the efficiency of this elegant procedure is a result of several favorable properties of Hamiltonian mechanics—namely volume preservation, reversibility, and conservation. Upon exact simulation of the Hamiltonian dynamics, these properties will lead to the acceptance probability of one, i.e., every proposed move will be accepted. In App. D, we will give intuition about these properties. Since the method developed later in §6 will subsume HMC as a special case, we delay the proof of correctness until then.

**Langevin Dynamics.** Langevin dynamics is a simplification of HMC, where only one leapfrog step is performed. Furthermore, what is usually referred to as Langevin dynamics is in fact the *uncorrected* Langevin dynamics, where the acceptance criterion is ignored, i.e., every proposed sample is accepted; see Algorithm 2. While uncorrected Langevin dynamics is guaranteed to converge when the energy function is smooth [31, §5.3], there is no such guarantee with the presence of a discontinuity in the energy function. Nevertheless, due to its simplicity, Langevin dynamics is the only gradient-based sampling method that has been applied for text generation applications.

### 5.2 Applications to Controlled Generation

One of the primary advantages of using gradient-based techniques for controlled generation is that it provides a means to sample from the conditional distribution Eq. (4) without having to calculate the normalization factor $Z_t$. For example in HMC algorithm (Algorithm 1), all we need to calculate regarding the potential energy $U(\mathbf{V}) = -\log p(\mathbf{V} \mid t)$ is two terms: (1) the gradient of $U$ with respect to $\mathbf{V}$: $\nabla U$, and (2) the difference between the potential energy of two points $\Delta U$ for the Metropolis criterion. Fortunately, both terms are independent of $Z_t$, and we can sample from the conditional distribution without the need to compute $Z_t$.

**MUCOLA.** As defined in Eq. (4), $p(\mathbf{V} \mid t)$ is $\mathbb{R}^d$-valued, so it is tempting to apply a gradient-based sampling technique to sample from it. And, indeed, Kumar et al. [21] have proposed such a scheme based on Langevin dynamics. They define the potential energy of the embeddings as

---

[7]There exist sophisticated methods [15] to tune the mass matrix, but given that LM gradients are expensive to compute, we will not attempt such methods in this paper.

$U(\mathbf{V}) \stackrel{\text{def}}{=} -\log p(\mathbf{V} \mid t)$ and apply Langevin dynamics out of the box; see Algorithm 3. However, since $U(\mathbf{V})$ is zero for any vector other than vectors in $\mathcal{B}$ (defined in §2.1), they modify the sampling process and project the proposed sample to the base embeddings set at each step of the sampling process (line 3 of the algorithm). The added projection step, however, is neither volume-preserving nor time-reversible. Hence, this sampling procedure does not sample from the intended distribution and is not a valid MCMC algorithm.

**COLD and other schemes.** Alternative approaches have been proposed in the literature to reformulate a language model as potential energy over the logit [COLD; 38] or simplex space [14, 20]. However, these formulations are not suitable for principled gradient-based sampling. COLD only employs a heuristic energy function to select among the candidate generations obtained via top-$k$ sampling, and the simplex-based approach requires an extra constraint to ensure the sample stays on the simplex.

# 6 Structured Voronoi Sampling

Given our structured Voronoi measure $p_{\mathrm{V}}$, one can apply HMC to sample from it. In this section, we take one step further and propose a variation of HMC that is more suitable to sample from $p_{\mathrm{V}}$. Importantly, $p_{\mathrm{V}}$ contains discontinuities whereas the generic leapfrog integrator does not account for such sudden jumps in the potential function. In other words, even if the leapfrog integrator itself is volume preserving and time-reversible, the sudden jumps in potential can lead to large deviations in the Hamiltonian value, causing a low acceptance rate. We therefore would like to find an alternative to leapfrog in such situations.

**A Physical Analogy.** In classical mechanics, discontinuity with smooth boundary in the potential function occurs naturally, e.g., in collision dynamics or a slab magnetic field, and is referred to as a potential barrier (or interface). Upon encountering a potential barrier, a particle will either be reflected from or transmitted through the barrier surface, depending on whether it has enough kinetic energy to overcome the potential jump (e.g., [8, §4.6.2]). Such behavior is similar to reflection–refraction phenomenon in optics. The key insight here is that, in both cases, the Hamiltonian is conserved.

**Reflection and Refraction.** To give a precise mechanical description of this behavior, suppose a particle encounters a potential barrier at position $\mathbf{x}$ with momentum $\mathbf{r}$. We can decompose momentum as $\mathbf{r} = \mathbf{r}_\perp + \mathbf{r}_\parallel$ where $\mathbf{r}_\perp$ is normal to the barrier and $\mathbf{r}_\parallel$ parallel to it. Let $\Delta U$ be the signed potential energy difference between the two sides of the barrier. If $\|\mathbf{r}_\perp\|_2^2 > 2\Delta U$, then the particle has enough kinetic energy to overcome the barrier, and its momentum's normal component will instantaneously become $\mathbf{r}'_\perp = \sqrt{\|\mathbf{r}_\perp\|_2^2 - 2\Delta U} \cdot \frac{\mathbf{r}_\perp}{\|\mathbf{r}_\perp\|_2}$ after being transmitted through the barrier (refraction). Otherwise, if $\|\mathbf{r}_\perp\|_2^2 \leq 2\Delta U$, the particle will be reflected from the barrier and the normal component will instantaneously become $\mathbf{r}'_\perp = -\mathbf{r}_\perp$. We show in App. E.1 that Hamiltonian is conserved in either case. The reflect–refract process is summarized in Algorithm 5.

## 6.1 A Sampling Algorithm

Noting that the discontinuity surfaces of $p_{\mathrm{V}}$ are all piecewise smooth, we can build on the above and develop a sampling algorithm for $p_{\mathrm{V}}$ to handle discontinuity in a principled and effective way. In fact, we only need to make one change to the generic HMC, which is updates $(\mathbf{x}, \mathbf{r})$ according to the mechanical description given above. Concretely, we need to replace step 2 in the HMC outline in §5.1: When a single step of leapfrog encounters no discontinuity, we may advance to the next point as in HMC; however, when there is discontinuity, if a full leapfrog step is taken, we need to proceed by repeatedly computing where the discontinuity is encountered, taking a smaller step up to the discontinuity and refracting–reflecting based on the potential energy difference. This process is continued until we have exhausted the step size. Since refracting–reflecting conserves Hamiltonian (App. E.1), this process yields a better acceptance rate in the presence of a discontinuity. See Algorithm 4 for the details of this sampling procedure, and App. F.1 for how to efficiently find discontinuities. We will supply proof of correctness in App. E.2.

**A note on calculating the base measure.** To adjust the momentum, one needs to compute $\Delta U$, which implies computing the difference between two base measures, as defined in Eq. (12). However, computing such an integral in a high-dimensional space is not practical. Therefore, make an

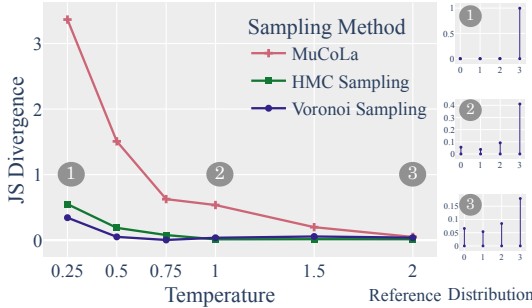

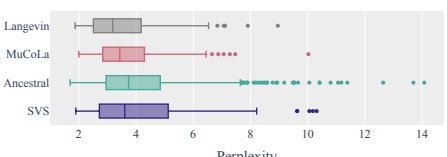

Figure 2: Left: JS divergence between the reference probability and empirical probability distribution. Voronoi Sampling clearly outperforms others in low temperatures. Right: reference probability distribution annealed with 3 temperatures: $0.25$ (peaked), $1$, and $2$ (close to uniform).

Figure 3: Perplexity of 100 samples taken with different gradient-based algorithms, compared to 1000 samples taken with the ancestral sampling (in green). While LANGEVIN, SVS, and MUCOLA are comparably close to the ancestral samples' distribution, SVS models the tail of the distribution better.

assumption that the base measures of Voronoi cells are equal, and thus do not have an effect on $\Delta U$. However, such an assumption might not hold. See App. A for limitations.

## 7  Experiments

We empirically assess the performance of Voronoi sampling in a series of experiments. In each experiment, we perform a grid search to find the best set of hyperparameters, these experimental details can be found in App. H. Our open-source implementation will be available upon publication.

### 7.1  Toy Example

We first apply our Voronoi sampling[8] method on the toy example discussed earlier in Example 1, where the reference probability distribution is tractable and known $p(\mathbf{x})$. The potential energy is then set to $U(\mathbf{x}) = -\log p_{\mathrm{V}}(\mathbf{x})$. Importantly, the toy experiment is intentionally designed such that the base measure of all of the Voronoi cells is equal, therefore, we can safely ignore calculating the base measure and arrive at exact sampling methods.

We compare Voronoi sampling to MUCOLA and HMC. To make a fair comparison, we add the Metropolis criterion[9] to the MUCOLA algorithm and only do one leapfrog step in all algorithms. Furthermore, to see the effect of the reference distribution on the performance of the sampling algorithm, we anneal this distribution with 6 temperatures, where the lower temperatures lead to peaky distributions and the higher temperatures to uniform-like distributions.

We take 200 samples after 500 burn-in iterations, and compare the empirical distributions of samples to the reference by measuring the Jensen–Shannon (JS) divergence between the two. As results in Fig. 2 show, the JS divergence between the reference distribution and the empirical distribution of Voronoi samples is the smallest. The difference between the methods is more pronounced at lower temperatures, as the change in potential energy is greater, resulting in more errors in the leapfrog integrator. In App. I we provide more empirical support that Voronoi sampling converges faster to the reference distribution, especially when we increase the dimensionality of the sampling space.

### 7.2  Sampling from Language Models

Next, we apply our method to sample from a language model. The underlying LM is a finetuned GPT-2[10] on E2E dataset [34]; see App. G for dataset statistics. As opposed to the previous

---

[8]Note that in the case of the toy experiment, we are not sampling a *sequence* of text, but rather a single embedding. To highlight this point, we use Voronoi sampling instead of *structured* Voronoi sampling to refer to our algorithm in this section.

[9]We accept transitioning from $\mathbf{x}^t$ to $\mathbf{x}^{t+1}$ with probability $e^{H^t - H^{t+1}}$.

[10]We use gpt2 checkpoint from the Huggingface library [47].

experiment, the reference distribution of the LM is not tractable. Therefore, we use the empirical distribution of ancestral samples as an unbiased estimate of the reference distribution. Note that ancestral sampling incrementally draws samples from a language model, where at each step of the generation a token $w_n$ is sampled with the probability given by the LM: $p(w_n \mid \boldsymbol{w}_{<n})$. Therefore, the process can give unbiased estimates of the reference distribution.

We follow §4 to define a structured Voronoi measure $p_{\mathrm{V}}$ using the LM probability. We then implement SVS, where the potential energy is set to $-\log p_{\mathrm{V}}$. To empirically measure the benefit of reflection–refraction step in SVS, we compare it to applying Langevin dynamics directly to $p_{\mathrm{V}}$. We implement MUCOLA as a baseline, which writes the potential energy using an embedding-augmented LM, i.e., Eq. (3).

We show the distribution of samples' perplexity in Fig. 3. The green trace is the empirical distribution of 1000 ancestral samples. While all the sampling methods result in distributions comparably close to ancestral samples' distribution, we observe that SVS manages to model the tail of the distribution better. On the other hand, MUCOLA and LANGEVIN tend to take samples from the mode of the distribution more often.

## 7.3 Controlled Generation

Finally, we apply our structured Voronoi sampling to 2 controlled generation task. The goal of the first task is to generate restaurant reviews for a target food type $t$, e.g., Italian, Fast food, Japanese, etc. The goal of the second task is to control the sentiment of the generations to enforce a positive sentiment. We train classifiers to predict the target $t$ (food type or positive sentiment) from the input sequence $p(t \mid \boldsymbol{w})$.[11] We implement two baselines:

**FUDGE.** Yang and Klein [48] offer a heuristic approach to sample from the conditional distribution. They incrementally sample tokens under the language model. At each sampling step, they adjust the probabilities given by the LM, by feeding each candidate prefix to a classifier and obtain the probability of that prefix following the control target.

**MUCOLA.** Kumar et al. [21] treat $p(\mathbf{V} \mid t)$, Eq. (4), as a distribution in $\mathbb{R}^d$ and apply Langevin dynamics directly to sample a sequence of embeddings $\mathbf{V}$. The potential energy is defined as $-\log p(\mathbf{V} \mid t)$. When rewriting this potential energy with Bayes' rule, it has been shown empirically, that adding a hyperparameter $\gamma$ is helpful to keep the balance between the classifier and LM. Therefore, the final potential energy is defined as:

$$U(\mathbf{V}) \stackrel{\text{def}}{=} -\log p(\mathbf{V}) - \gamma \log p(t \mid \mathbf{V}). \tag{17}$$

As mentioned earlier, $p(\mathbf{V} \mid t)$ only places a positive probability on a countable set. We, therefore, use Def. 3 to define structured Voronoi measures and set the potential energy to

$$U(\mathbf{x}) \stackrel{\text{def}}{=} -\log p_{\mathrm{V}}(\mathbf{x}) - \gamma \log p_{\mathrm{V}}(t \mid \mathbf{x}). \tag{18}$$

We then apply Langevin dynamics and SVS to sample according to this potential energy.

**Evaluation.** We sample 120 sentences of length 20[12] and evaluate the generations on three metrics:

- **Success:** is defined as the percentage of generations that adhere to the control target. To determine whether a generation conforms to the specified target we use an *evaluator classifier*.
- **Fluency:** is measured by the mean and standard deviation of perplexity under the language model.
- **Diversity:** is measured by the mean number of distinct $n$-grams ($n = 1, 2, 3$) in a set of samples, normalized by the length of the sequence.

As results in Table 1 show,[13] FUDGE tends to achieve the highest diversity, however, it fails to follow the control target. MUCOLA either generates fluent results without paying enough attention to the control, or sacrifices fluency in favor of following the control target; thus, the high variance in success rates. Both LANGEVIN and SVS result in a high success rate and maintain fluency and diversity, and SVS is effective in maintaining a balance between various metrics and producing fluent sentences that adhere to control targets.

---

[11]See App. H for more experimental details about classifiers.

[12]We sample 20 sentences per control target.

[13]Please refer to Fig. 7 to see a visualization of the topic control results, and to Table 6 for results per target type.

| | Topic Control | | | | | Sentiment Control | | | |
|---|---|---|---|---|---|---|---|---|---|
| | Success(↑) | PPL(↓) | Dist-1(↑) | Dist-2(↑) | Dist-3(↑) | Success(↑) | PPL(↓) | Dist-1(↑) | Dist-2(↑) | Dist-3(↑) |
| GPT2 | $0.12 \pm 0.10$ | $5.10 \pm 2.06$ | 0.40 | 0.56 | 0.67 | $0.55 \pm 0.49$ | $21.33 \pm 27.17$ | 0.40 | 0.60 | 0.71 |
| FUDGE | $0.30 \pm 0.12$ | $5.59 \pm 0.60$ | 0.39 | 0.55 | 0.65 | $0.57 \pm 0.49$ | $24.27 \pm 12.46$ | 0.40 | 0.60 | 0.70 |
| MUCOLA | $0.58 \pm 0.23$ | $33.09 \pm 36.32$ | 0.26 | 0.40 | 0.51 | $0.66 \pm 0.47$ | $85.74 \pm 152.18$ | 0.28 | 0.42 | 0.53 |
| LANGEVIN | $0.91 \pm 0.12$ | $14.26 \pm 2.55$ | 0.24 | 0.39 | 0.51 | $0.82 \pm 0.38$ | $26.76 \pm 14.42$ | 0.16 | 0.30 | 0.41 |
| SVS | $0.92 \pm 0.05$ | $13.9 \pm 2.04$ | 0.22 | 0.37 | 0.49 | $0.84 \pm 0.36$ | $32.73 \pm 16.70$ | 0.14 | 0.28 | 0.41 |

Table 1: Evaluation of different sampling methods on controlled generation, using three criteria: success in following the control target (measured by the evaluator classifier), fluency (measured by perplexity), and diversity.

## 8 Related Work

**Controlled Generation.** Numerous approaches have been proposed to enforce controls during the text generation process [19, 24, 41, *inter alia*]. For example, weighted decoding [10, 17] scores each candidate token with a weighted sum of its score under the language model and its adherence to control targets, subsequently selecting candidates with the highest scores. FUDGE method adopts a similar scoring function, resembling a Bayesian formulation for $p(\boldsymbol{w} \mid t)$. After making simplifying assumptions and factorizing $p(\boldsymbol{w} \mid t)$, FUDGE samples tokens autoregressively based on their scores. More recently, a line of research attempts to directly sample from $p(\boldsymbol{w} \mid t)$ by reformulating it as an energy-based model and sampling from it using efficient gradient-based sampling algorithms. As discussed in §5.2 COLD reformulates $p(\boldsymbol{w} \mid t)$ as an energy-based model on the logit space and uses that to select samples with high energy from a number of candidate generations. MUCOLA offers a sampling algorithm motivated by Langevin Dynamics that operates in the embedding space.

**Gradient-based Sampling.** Our work is closely related to the line of research that makes use of gradient information to sample from complex distributions [7, 31, 46]. Gradient-based samplers [15, 32] are shown to be highly effective when sampling from continuous distributions [3, 4, 37]. However, it is a difficult problem to adapt gradient-based samplers to discrete settings [36, 50]. More recently, several papers proposed promising gradient-based MCMC for discrete distribution that are reversible chains [11, 39, 49]. Our work instead formulates an irreversible Markov chain based on HMC. We leave it to future work to explore the utility of these recent papers on text generation.

## 9 Conclusion

In this work, we propose structured Voronoi sampling, a principled gradient-based sampling method for text generation. To formulate the energy function used in SVS, we define structured Voronoi measures on the embedding space and show how such measures can encode language models. In a controlled generation task, SVS outperformed other sampling methods in following the control target while producing comparably fluent and diverse samples.

## Broader Impacts

It has been repeatedly shown that LMs can generate harmful, toxic, or non-factual content [9, 35, 42]. In fact, an application of the controlled generation scheme discussed in this paper could be to mitigate such issues. However, the same method could be used to generate misinformation, or toxic content intentionally.

## Acknowledgements

We thank Tim Vieira, Tiago Pimentel, and Clara Meister for their feedback on this work. We also thank the anonymous reviewers for their constructive feedback during the review process. Afra Amini is supported by ETH AI Center doctoral fellowship.

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

# A   Limitations

**Approximating the base measure.**   In this work, we go one step closer to implementing a principled approach for text generation. However, in our text generation experiments, we follow the previous work in using uncorrected Langevin dynamics. Moreover, when implementing SVS for text generation, we assume all Voronoi cells to have an equal base measure, which might not hold.

**Efficiency.**   As reported in App. H, all gradient-based samplers are considerably slower than ancestral sampling and heuristics such as FUDGE. Further efforts are needed to make gradient-based methods faster.

**Text Quality.**   As shown in previous work, sequences with high probability under the LM can be repetitive, dull, or degenerate [16]. Therefore, as with any other sampling method, SVS might sample degenerate sentences, and the quality of the samples depends on the LM probability distribution. This is an active area of research with various proposals from changing the loss function during training [45], to modifying the decoding objective [28].

# B   A Note on the Choice of the Gaussian Measure

We thank the reviewers and the meta reviewer for their careful assessment of this work and their thoughtful feedback. As a response to an important point raised by the meta reviewer, here we explain the reasoning behind using a Gaussian measure in Eq. (12). The gradient $g_m$ is only defined at Voronoi centers and the goal is to follow a direction similar to $g_m$ when we are in the vicinity of the Voronoi center, i.e., in the Voronoi cell. We note that this direction needs to be adjusted depending on the position in the Voronoi cell $\mathbf{x}$. A natural choice for this would be to use a truncated Gaussian that is centered at $g_m$, which implies that the gradient in $\mathbf{x}$ should be $g_m - \mathbf{x}$, to follow a similar direction $g_m$ at the center of the Voronoi.

# C   Proofs

## C.1   Proof of Proposition 1

**Proposition 1.** *Let $p = [p_1, \ldots, p_M]$ be an embedding-augmented distribution with embeddings $\{\mathbf{v}_m\}_{m=1}^M \subset \mathbb{R}^d$, and let $p_V$ be the corresponding Voronoi measure Eq. (7). Then, $p_V(C_m) = p_m$ where $C_m$ is defined as in Eq. (6). See App. C.1 for proof.*

*Proof.*

$$p_V(C_m) = \int_{C_m} p_V(\mathbf{x}) \mathrm{d}\lambda \tag{19a}$$

$$= \int_{C_m} \frac{p_m}{\mu(C_m)} \frac{\mathrm{d}\mu}{\mathrm{d}\lambda}(\mathbf{x}) \mathrm{d}\lambda \tag{19b}$$

$$= \frac{p_m}{\mu(C_m)} \int_{C_m} \frac{\mathrm{d}\mu}{\mathrm{d}\lambda}(\mathbf{x}) \mathrm{d}\lambda \tag{19c}$$

$$= \frac{p_m}{\mu(C_m)} \mu(C_m) = p_m \tag{19d}$$

∎

## C.2   Proof of Proposition 2

**Proposition 2.** *Let $\mu$ be a measure on $\mathbb{R}^d$. Then, we have the product measure space as $\mu(C_{\boldsymbol{m}}) = \prod_{n=1}^N \mu(C_{m_n})$. See App. C.2 for proof.*

*Proof.* Let $\boldsymbol{m} = [m_1, \ldots, m_N] \in [M]^N$. We have

$$\mu(C_{\boldsymbol{m}}) = \mu\left(\prod_{n=1}^{N} C_{m_n}\right) \tag{20a}$$

$$= \prod_{n=1}^{N} \mu(C_{m_n}) \tag{20b}$$

∎

## C.3 Proof of Proposition 3

**Proposition 3.** *Eq. (13) is absolutely continuous with respect to the Lebesgue measure $\lambda$. See App. C.3 for proof.*

*Proof.* Choose $E \in \mathscr{B}(\mathbb{R}^d)$ such that $\lambda(E) = 0$.[14] Note that $E = (E \setminus \mathcal{K}) \cup \bigcup_{m=1}^{M}(E \cap C_m)$ where $\mu(E \setminus \mathcal{K}) = 0$ by Eq. (13). Since the Gaussian measure over $\mathbb{R}^d$ itself is absolutely continuous with respect to $\lambda$, we can also conclude that $\mu(E \cap C_m) = 0$ for any $m$. Hence, $\mu(E) \leq \mu(E \setminus \mathcal{K}) + \sum_m \mu(E \cap C_m) = 0$ which means $\mu(E) = 0$. So $\mu \ll \lambda$. ∎

## C.4 Proof of Proposition 4

**Proposition 4.** *The gradient of the log of the Voronoi measure $p_V$ is given by*

$$\nabla_{\mathbf{x}} \log p_V(\mathbf{x}) = \begin{cases} \mathbf{g}_{\boldsymbol{m}} - \mathbf{x}, & \textit{if } \mathbf{x} \in \text{int}(C_{\boldsymbol{m}}) \\ \textit{undefined}, & \textit{if } \mathbf{x} \in \partial C_{\boldsymbol{m}} \\ \mathbf{0}, & \textit{otherwise} \end{cases} \tag{14}$$

*where the first two blocks in the case statement apply if there exists some $\boldsymbol{m}$ such that $\mathbf{x} \in \text{int}(C_{\boldsymbol{m}})$ or $\mathbf{x} \in \partial C_{\boldsymbol{m}}$. See App. C.4 for proof.*

*Proof.* We have three cases. Suppose $\mathbf{x} \in \text{int}(C_m)$ for some $m \in [M]$. Then, by Observation 1, we have that $p_V(\mathbf{x})$, and, thus, $\log p_V(\mathbf{x})$ is differentiable. Direct computation reveals:

$$\nabla_{\mathbf{x}} \log p_V(\mathbf{x}) = \underbrace{\nabla_{\mathbf{x}} \log \frac{p_m}{\mu(C_m)}}_{=0} + \nabla_{\mathbf{x}} \log \frac{\mathrm{d}\mu}{\mathrm{d}\lambda} \tag{21a}$$

$$= \nabla_{\mathbf{x}} \log \exp\left(-\frac{1}{2}\|\mathbf{g}_m - \mathbf{x}\|_2^2\right) - \underbrace{\nabla_{\mathbf{x}} \log \mu(C_m)}_{=0}$$

$$= -\frac{1}{2}\nabla_{\mathbf{x}}\|\mathbf{g}_m - \mathbf{x}\|_2^2 \tag{21b}$$

$$= \mathbf{g}_m - \mathbf{x} \tag{21c}$$

Next, suppose $\mathbf{x} \notin \bigcup_{m=1}^{M} C_m$. Then, the measure is zero, so the gradient is as well. Finally, we have that $\mathbf{x} \in \partial C_m$ for some $m$. Then, by the next Observation (Observation 1), we have that $p_V$ is discontinuous, so the derivative is not defined. ∎

**Observation 1.** *A $(\mathcal{K}, \mu)$-Voronoi measure $p_V$ is differentiable with respect to $p_m$ on the set $\cup_{m=1}^{M}\text{int}(C_m)$, and discontinuous on the set $\mathcal{K} \setminus \cup_{m=1}^{M}\text{int}(C_m)$, i.e., the union of the Voronoi cells' boundaries $\cup_{m=1}^{M}\partial C_m$.*

---

[14] We use $\mathscr{B}(\cdot)$ to denote the standard Borel $\sigma$-algebra.

## D  Properties of Hamiltonian Dynamics

**Preservation of Measure.**  First of all, the Hamiltonian equations are volume- or measure-preserving. Intuitively, this means that if we move a region in the phase space along the dynamics for an arbitrary amount of time, the volume of the region would stay unchanged.[15] Concretely, we can define a function

$$g^t : (\mathbf{x}(0), \mathbf{r}(0)) \mapsto (\mathbf{x}(t), \mathbf{r}(t)) \tag{22}$$

as moving every point in the phase space along the Hamiltonian equations for some time $t$.[16] Then, using $g^t$ as a change of variable would leave the underlying probability measure unchanged.

**Time Reversibility.**  Next, the Hamiltonian equations are time reversible, meaning that if the equations can move $(\mathbf{x}, \mathbf{r})$ to $(\mathbf{x}', \mathbf{r}')$, then it would also take $(\mathbf{x}', -\mathbf{r}')$ to $(\mathbf{x}, -\mathbf{r})$. Due to our choice of distribution of $\mathbf{r}$ is always symmetric, these properties simplify the acceptance probability to be only the energy difference, i.e., we can ignore the conditional probabilities in the standard Metropolis–Hastings acceptance probability.

**Conservation of Hamiltonian.**  Finally, we can quickly verify that the Hamiltonian $H$ is conserved by the Hamiltonian dynamics:[17]

$$\frac{\mathrm{d}H}{\mathrm{d}t} = \sum_i \frac{\partial H}{\partial \mathbf{x}_i} \frac{\mathrm{d}\mathbf{x}_i}{\mathrm{d}t} + \sum_i \frac{\partial H}{\partial \mathbf{r}_i} \frac{\mathrm{d}\mathbf{r}_i}{\mathrm{d}t} \tag{23}$$

$$= \sum_i \frac{\partial H}{\partial \mathbf{x}_i} \frac{\partial H}{\partial \mathbf{r}_i} - \sum_i \frac{\partial H}{\partial \mathbf{r}_i} \frac{\partial H}{\partial \mathbf{x}_i} \qquad \text{(definition in Eq. (16))} \tag{24}$$

$$= \sum_i \frac{\partial H}{\partial \mathbf{x}_i} \frac{\partial H}{\partial \mathbf{r}_i} - \sum_i \frac{\partial H}{\partial \mathbf{x}_i} \frac{\partial H}{\partial \mathbf{r}_i} \qquad \text{(symmetry)} \tag{25}$$

$$= 0 \tag{26}$$

This shows that the acceptance probability will be exactly 1 if the Hamiltonian equations are integrated exactly. In practice, numerical errors will lead to fluctuations in $H$. The Metropolis–Hastings acceptance probability ensures detailed balance.

## E  Details of Voronoi Sampling

### E.1  Conservation of Hamiltonian in Refraction–Reflection

**Proposition 5.** *A single step of refraction–reflection conserves the Hamiltonian.*

*Proof.*  Suppose an instantaneous refraction–reflection takes $(\mathbf{x}, \mathbf{r})$ to $(\mathbf{x}', \mathbf{r}')$. In other words, $(\mathbf{x}, \mathbf{r})$ is the particle's coordinate and momentum at the boundary prior to the refraction–reflection, and $(\mathbf{x}', \mathbf{r}')$ is the particle's coordinate and momentum after the refraction–reflection, which includes instantaneous changes in its potential energy and momentum. Recall the refraction–reflection equation is

$$\mathbf{r}'_\perp = \begin{cases} \sqrt{\|\mathbf{r}_\perp\|_2^2 - 2\Delta U} \cdot \frac{\mathbf{r}_\perp}{\|\mathbf{r}_\perp\|_2^2} & \text{(refraction when } \|\mathbf{r}_\perp\|_2^2 > 2\Delta U \text{ )} \\ -\mathbf{r}_\perp & \text{(reflection when } \|\mathbf{r}_\perp\|_2^2 \leq 2\Delta U \text{ )} \end{cases} \tag{27a}$$

---

[15]This result is known as Liouville's theorem in classical mechanics [2, §16]. Upon noticing that the Hamiltonian dynamics is divergenceless, one can interpret it as a simple application of the (higher-dimensional) divergence theorem, which itself is a consequence of the generalized Stokes' theorem.

[17]We say a transformation taking $(\mathbf{x}, \mathbf{r})$ to $(\mathbf{x}', \mathbf{r}')$ **conserves** Hamiltonian if $H(\mathbf{x}, \mathbf{r}) = H(\mathbf{x}', \mathbf{r}')$.

[17]This function is called the Hamiltonian phase flow, or simply Hamiltonian flow [2, §16]. Flow is a general and important notion in the study of differential equations and related subjects, in which some additive group acts on $(\mathbb{R}, +)$. In the case of a smooth Hamiltonian $H$, $\{g^t\}_{t \in \mathbb{R}}$ can be seen a one-parameter group of diffeomorphisms over $\mathbb{R}^{2d}$.

where $\mathbf{r} = \mathbf{r}_\perp + \mathbf{r}_\parallel$ is the normal decomposition of $\mathbf{r}$ and $\mathbf{r}' = \mathbf{r}'_\perp + \mathbf{r}'_\parallel$. Then,

$$H(\mathbf{x}', \mathbf{r}') = U(\mathbf{x}') + \frac{1}{2}\|\mathbf{r}'\|_2^2 \tag{27b}$$

$$= U(\mathbf{x}') + \frac{1}{2}\|\mathbf{r}'_\perp + \mathbf{r}'_\parallel\|_2^2 \tag{27c}$$

$$= U(\mathbf{x}') + \frac{1}{2}\|\mathbf{r}'_\perp\|_2^2 + \langle \mathbf{r}_\perp, \mathbf{r}_\parallel \rangle + \frac{1}{2}\|\mathbf{r}'_\parallel\|_2^2 \tag{27d}$$

$$= U(\mathbf{x}') + \frac{1}{2}\|\mathbf{r}'_\perp\|_2^2 + \frac{1}{2}\|\mathbf{r}'_\parallel\|_2^2 \qquad ( \langle \mathbf{r}_\perp, \mathbf{r}_\parallel \rangle = 0) \tag{27e}$$

$$= \begin{cases} U(\mathbf{x}) + \Delta U + \frac{1}{2}(\|\mathbf{r}_\perp\|_2^2 - 2\Delta U) + \frac{1}{2}\|\mathbf{r}'_\parallel\|_2^2 & \text{(refraction)} \\ U(\mathbf{x}) + \frac{1}{2}\| - \mathbf{r}_\perp\|_2^2 + \frac{1}{2}\|\mathbf{r}_\parallel\|_2^2 & \text{(reflection)} \end{cases} \tag{27f}$$

$$= \begin{cases} U(\mathbf{x}) + \cancel{\Delta U} + \frac{1}{2}\|\mathbf{r}_\perp\|_2^2 - \cancel{\Delta U} + \frac{1}{2}\|\mathbf{r}'_\parallel\|_2^2 & \text{(refraction)} \\ U(\mathbf{x}) + \frac{1}{2}\| - \mathbf{r}_\perp\|_2^2 + \frac{1}{2}\|\mathbf{r}_\parallel\|_2^2 & \text{(reflection)} \end{cases} \tag{27g}$$

$$= U(\mathbf{x}) + \frac{1}{2}\|\mathbf{r}\|_2^2 \tag{27h}$$

$$= H(\mathbf{x}, \mathbf{r}). \tag{27i}$$

∎

## E.2 Proof of Correctness

In this section, we give a proof of correctness of our sampler by establishing its detailed balance. We will first prove several useful properties of the sampler in E.2.1 to E.2.3 and then combine them to prove the detailed balance in App. E.2.4.

### E.2.1 Measure Preservation in Leapfrog

First of all, we note that the two leapfrog steps are measure-preserving, since they are composed of a series of *shear* transformations.

**Proposition 6.** *The leapfrog steps are measure-preserving.*

*Proof.* Recall that the two leapfrog steps used in HMC (algorithm 1) are $(\mathbf{x}, \mathbf{r}) \to (\mathbf{r}, \mathbf{r} - \frac{\varepsilon}{2}\nabla U(\mathbf{x}))$ and $(\mathbf{x}, \mathbf{r}) \to (\mathbf{x} + \varepsilon \mathbf{r}, \mathbf{r})$.

By the change-of-variable formula, to show a transformation is measure-preserving, it suffices to show that its Jacobian has determinant 1. The leapfrog step $(\mathbf{x}, \mathbf{r}) \to (\mathbf{x}, \mathbf{r} - \frac{\varepsilon}{2}\nabla U(\mathbf{x}))$ as a function has the Jacobian

$$\frac{\partial(\mathbf{x}, \mathbf{r} - \frac{\varepsilon}{2}\nabla U(\mathbf{x}))}{\partial(\mathbf{x}, \mathbf{r})} = \begin{pmatrix} I & 0 \\ -\frac{\varepsilon}{2}\nabla^2 U(\mathbf{x}) & I \end{pmatrix} \tag{28}$$

where $\nabla^2 U(\mathbf{x})$ denotes the Hessian of $U$. This Jacobian (28) has determinant 1 and hence the leapfrog transformation $(\mathbf{x}, \mathbf{r})$ to $(\mathbf{x}, \mathbf{r} - \frac{\varepsilon}{2}\nabla U(\mathbf{x}))$ is measure preserving. Similarly, the other leapfrog step $(\mathbf{x}, \mathbf{r}) \to (\mathbf{x} + \varepsilon \mathbf{r}, \mathbf{r})$ has Jacobian

$$\frac{\partial(\mathbf{x} + \varepsilon \mathbf{r}, \mathbf{r})}{\partial(\mathbf{x}, \mathbf{r})} = \begin{pmatrix} I & \varepsilon I \\ 0 & I \end{pmatrix} \tag{29}$$

which also has determinant 1. ∎

### E.2.2 Measure Preservation in Refraction–Reflection

The computation for showing that refraction and reflection steps are measure preserving is slightly more involved than the leapfrog case. Our proof below largely follows the one found in Mohasel Afshar and Domke [30], where we simplified and corrected some details.

**Proposition 7.** *The refraction–reflection step is measure-preserving.*

*Proof.* We consider a single step of refraction–reflection as a function $\gamma$ that takes $(\mathbf{x}, \mathbf{r})$ to $(\mathbf{x}', \mathbf{r}')$ with total step size $\varepsilon$ and with only one discontinuity. Note that, without loss of generality, we may only consider the discontinuity surface to be the hyperplane $\mathbf{x}_1 = 0$ since it is equivalent to any tangent hyperplane of the discontinuity (hyper)surface through an affine transformation. Then, a normal vector to the hyperplane $\mathbf{x}_1 = 0$ is $\mathbf{e}_1 = (1, 0, \ldots, 0)$ and

$$\mathbf{r}_\perp = (\mathbf{r}_1, 0, \ldots, 0) \quad \text{and} \quad \mathbf{r}_\| = (0, \mathbf{r}_2, \ldots, \mathbf{r}_d). \tag{30}$$

In this case,

$$\forall\, i \geq 2, \quad \mathbf{r}'_i = \mathbf{r}_i \quad \text{and} \quad \mathbf{x}'_i = \mathbf{x}_i + \varepsilon \mathbf{r}_i \tag{31}$$

since $\mathbf{r}_\|$ is not affected by refraction–reflection. Therefore,

$$\forall\, i \geq 2, \quad \frac{\partial \mathbf{x}'_i}{\partial \mathbf{x}_i} = \frac{\partial \mathbf{r}'_i}{\partial \mathbf{r}_i} = 1 \tag{32}$$

and

$$\forall\, i, j \geq 2 \text{ and } j \neq i, \quad \frac{\partial \mathbf{x}'_i}{\partial \mathbf{x}_j} = \frac{\partial \mathbf{x}'_i}{\partial \mathbf{r}_j} = \frac{\partial \mathbf{r}'_i}{\partial \mathbf{x}_j} = \frac{\partial \mathbf{r}'_i}{\partial \mathbf{r}_j} = 0. \quad \text{(by Eq. (31))} \tag{33}$$

Eq. (32) and Eq. (33) shows that the absolute determinant of the Jacobian of $\gamma$ is the same as

$$|\det \nabla \gamma| = \left| \det \begin{pmatrix} \frac{\partial \mathbf{x}'_1}{\partial \mathbf{x}_1} & \frac{\partial \mathbf{x}'_1}{\partial \mathbf{r}_1} \\ \frac{\partial \mathbf{r}'_1}{\partial \mathbf{x}_1} & \frac{\partial \mathbf{r}'_1}{\partial \mathbf{r}_1} \end{pmatrix} \right| = \left| \frac{\partial \mathbf{x}'_1}{\partial \mathbf{x}_1} \frac{\partial \mathbf{r}'_1}{\partial \mathbf{r}_1} - \frac{\partial \mathbf{x}'_1}{\partial \mathbf{r}_1} \frac{\partial \mathbf{r}'_1}{\partial \mathbf{x}_1} \right|. \tag{34}$$

We analyze all four quantities in Eq. (34) individually below. Again, without loss of generality, suppose $\mathbf{x}_1 \leq 0$ and $\mathbf{x}'_1 \geq 0$. Let $\Delta U(\mathbf{x})$ be the function of the signed potential difference as $\mathbf{x}_1$ changes from negative to positive, defined only when $\mathbf{x}_1 = 0$. Let $\mathbf{y} = (0, \mathbf{y}_2, \ldots, \mathbf{y}_d)$ be the point where the discontinuity is encountered. In other words, the particle begins at $\mathbf{x}$, moved past $\mathbf{y}$ and arrived at $\mathbf{x}'$.

**Refraction.** In the case of refraction, $\mathbf{r}_1^2 > 2\Delta U(\mathbf{y})$ and

$$\mathbf{r}'_1 = \sqrt{\mathbf{r}_1^2 - 2\Delta U(\mathbf{y})}. \tag{35}$$

Let $\delta$ be the step size it takes to reach $\mathbf{y}$. Then

$$\delta = -\frac{\mathbf{x}_1}{\mathbf{r}_1}, \tag{36}$$

$$\mathbf{x}'_1 = (\varepsilon - \delta)\mathbf{r}'_1 = \left( \varepsilon + \frac{\mathbf{x}_1}{\mathbf{r}_1} \right) \mathbf{r}'_1, \tag{37}$$

and

$$\mathbf{y} = \mathbf{x} + \delta \mathbf{r} = \mathbf{x} - \frac{\mathbf{x}_1}{\mathbf{r}_1} \mathbf{r}. \tag{38}$$

Then,

$$\frac{\partial \mathbf{x}'_1}{\partial \mathbf{x}_1} = \frac{\partial}{\partial \mathbf{x}_1} \left( \varepsilon + \frac{\mathbf{x}_1}{\mathbf{r}_1} \right) \mathbf{r}'_1 \qquad \text{(Eq. (37))} \tag{39a}$$

$$= \varepsilon \frac{\partial \mathbf{r}'_1}{\partial \mathbf{x}_1} + \frac{1}{\mathbf{r}_1} \mathbf{r}'_1 + \frac{\mathbf{x}_1}{\mathbf{r}_1} \frac{\partial \mathbf{r}'_1}{\partial \mathbf{x}_1} \qquad \text{(product rule)} \tag{39b}$$

$$= \left( \varepsilon + \frac{\mathbf{x}_1}{\mathbf{r}_1} \right) \frac{\partial \mathbf{r}'_1}{\partial \mathbf{x}_1} + \frac{\mathbf{r}'_1}{\mathbf{r}_1} \qquad \text{(product rule)}, \tag{39c}$$

$$\frac{\partial \mathbf{x}'_1}{\partial \mathbf{r}_1} = \frac{\partial}{\partial \mathbf{r}_1} \left( \varepsilon + \frac{\mathbf{x}_1}{\mathbf{r}_1} \right) \mathbf{r}'_1 \qquad \text{(Eq. (37))} \tag{40a}$$

$$= -\frac{\mathbf{x}_1 \mathbf{r}'_1}{\mathbf{r}_1^2} + \left( \varepsilon + \frac{\mathbf{x}_1}{\mathbf{r}_1} \right) \frac{\partial \mathbf{r}'_1}{\partial \mathbf{r}_1}, \tag{40b}$$

$$\frac{\partial \mathbf{r}_1'}{\partial \mathbf{x}_1} = \frac{\partial \sqrt{\mathbf{r}_1^2 - 2\Delta U(\mathbf{y})}}{\partial \mathbf{x}_1} \qquad \text{(by Eq. (35))} \qquad (41a)$$

$$= \frac{1}{2\sqrt{\mathbf{r}_1^2 - 2\Delta U(\mathbf{y})}} \frac{\partial (\mathbf{r}_1^2 - 2\Delta U(\mathbf{y}))}{\partial \mathbf{x}_1} \qquad \text{(chain rule)} \qquad (41b)$$

$$= -\frac{1}{\mathbf{r}_1'} \frac{\partial \Delta U(\mathbf{y})}{\partial \mathbf{x}_1}, \qquad (41c)$$

and

$$\frac{\partial \mathbf{r}_1'}{\partial \mathbf{r}_1} = \frac{\partial \sqrt{\mathbf{r}_1^2 - 2\Delta U(\mathbf{y})}}{\partial \mathbf{r}_1} \qquad \text{(by Eq. (35))} \qquad (42a)$$

$$= \frac{1}{2\sqrt{\mathbf{r}_1^2 - 2\Delta U(\mathbf{y})}} \frac{\partial (\mathbf{r}_1^2 - 2\Delta U(\mathbf{y}))}{\partial \mathbf{r}_1} \qquad \text{(chain rule)} \qquad (42b)$$

$$= \frac{\mathbf{r}_1}{\mathbf{r}_1'} - \frac{1}{\mathbf{r}_1'} \frac{\partial \Delta U(\mathbf{y})}{\partial \mathbf{r}_1}. \qquad (42c)$$

Then,

$$\det \nabla \gamma = \frac{\partial \mathbf{x}_1'}{\partial \mathbf{x}_1} \frac{\partial \mathbf{r}_1'}{\partial \mathbf{r}_1} - \frac{\partial \mathbf{x}_1'}{\partial \mathbf{r}_1} \frac{\partial \mathbf{r}_1'}{\partial \mathbf{x}_1} \qquad \text{(Eq. (34))}$$

$$\qquad (43a)$$

$$= \left( \left( \varepsilon + \frac{\mathbf{x}_1}{\mathbf{r}_1} \right) \frac{\partial \mathbf{r}_1'}{\partial \mathbf{x}_1} + \frac{\mathbf{r}_1'}{\mathbf{r}_1} \right) \frac{\partial \mathbf{r}_1'}{\partial \mathbf{r}_1} - \left( \frac{\mathbf{x}_1 \mathbf{r}_1'}{\mathbf{r}_1^2} + \left( \varepsilon + \frac{\mathbf{x}_1}{\mathbf{r}_1} \right) \frac{\partial \mathbf{r}_1'}{\partial \mathbf{r}_1} \right) \frac{\partial \mathbf{r}_1'}{\partial \mathbf{x}_1} \quad \text{(Eq. (39) and (40))}$$

$$\qquad (43b)$$

$$= \frac{\mathbf{r}_1'}{\mathbf{r}_1} \left( \frac{\partial \mathbf{r}_1'}{\partial \mathbf{r}_1} + \frac{\mathbf{x}_1}{\mathbf{r}_1} \frac{\partial \mathbf{r}_1'}{\partial \mathbf{x}_1} \right) \qquad (43c)$$

$$= \frac{\mathbf{r}_1'}{\mathbf{r}_1} \left( \frac{\mathbf{r}_1}{\mathbf{r}_1'} - \frac{1}{\mathbf{r}_1'} \frac{\partial \Delta U(\mathbf{y})}{\partial \mathbf{r}_1} - \frac{\mathbf{x}_1}{\mathbf{r}_1} \frac{1}{\mathbf{r}_1'} \frac{\partial \Delta U(\mathbf{y})}{\partial \mathbf{x}_1} \right) \qquad \text{(Eq. (41) and (42))}$$

$$\qquad (43d)$$

$$= 1 - \frac{1}{\mathbf{r}_1} \left( \frac{\partial \Delta U(\mathbf{y})}{\partial \mathbf{r}_1} + \frac{\mathbf{x}_1}{\mathbf{r}_1} \frac{\partial \Delta U(\mathbf{y})}{\partial \mathbf{x}_1} \right) \qquad (43e)$$

where

$$\frac{\partial \Delta U(\mathbf{y})}{\partial \mathbf{r}_1} + \frac{\mathbf{x}_1}{\mathbf{r}_1} \frac{\partial \Delta U(\mathbf{y})}{\partial \mathbf{x}_1} \qquad (44a)$$

$$= \sum_i \frac{\partial \Delta U(\mathbf{y})}{\partial \mathbf{y}_i} \frac{\partial \mathbf{y}_i}{\partial \mathbf{r}_1} + \frac{\mathbf{x}_1}{\mathbf{r}_1} \sum_i \frac{\partial \Delta U(\mathbf{y})}{\partial \mathbf{y}_i} \frac{\partial \mathbf{y}_i}{\partial \mathbf{x}_1} \qquad \text{(chain rule)} \qquad (44b)$$

$$= \sum_i \frac{\partial \Delta U(\mathbf{y})}{\partial \mathbf{y}_i} \frac{\mathbf{x}_1 \mathbf{r}_i}{\mathbf{r}_1^2} - \frac{\mathbf{x}_1}{\mathbf{r}_1} \sum_i \frac{\partial \Delta U(\mathbf{y})}{\partial \mathbf{y}_i} \frac{\mathbf{r}_i}{\mathbf{r}_1} \qquad \text{(Eq. (38))} \qquad (44c)$$

$$= 0. \qquad (44d)$$

Hence, Eq. (43) and Eq. (44) together imply that

$$|\det \nabla \gamma| = 1. \qquad (45)$$

So $\gamma$ is measure-preserving in the case of refraction.

**Reflection.** In the case of reflection, $\mathbf{r}_1^2 \leq 2\Delta U(\mathbf{y})$ and

$$\mathbf{r}_1' = -\mathbf{r}_1. \qquad (46)$$

As in the case of refraction, let $\delta$ be the step size it takes to reach $\mathbf{y}$. Then

$$\delta = -\frac{\mathbf{x}_1}{\mathbf{r}_1}, \qquad (47)$$

$$\mathbf{x}_1' = (\varepsilon - \delta)\mathbf{r}_1' = \left( \varepsilon + \frac{\mathbf{x}_1}{\mathbf{r}_1} \right)(-\mathbf{r}_1') = -\varepsilon \mathbf{r}_1' - \mathbf{x}_1. \qquad (48)$$

We can then directly calculate

$$\frac{\partial \mathbf{x}_1'}{\partial \mathbf{x}_1} = -1, \quad \frac{\partial \mathbf{x}_1'}{\partial \mathbf{x}_1} = -\varepsilon, \qquad \text{(Eq. (48))} \qquad (49a)$$

$$\frac{\partial \mathbf{x}_1'}{\partial \mathbf{x}_1} = 0, \quad \frac{\partial \mathbf{x}_1'}{\partial \mathbf{x}_1} = -1. \qquad \text{(Eq. (46))} \qquad (49b)$$

Hence,

$$|\det \nabla \gamma| = |(-1)(-1) - 0 \cdot (-\varepsilon)| = 1. \qquad (50)$$

So $\gamma$ is measure preserving in the case of reflection as well.

∎

### E.2.3 Time Reversibility

**Proposition 8.** *The refraction–reflection step is time-reversible.*

*Proof.* A refraction–reflection step is time-reversible means that, if a single step of refraction–reflection procedure takes $(\mathbf{x}, \mathbf{r})$ to $(\mathbf{x}', \mathbf{r}')$, then it would also take $(\mathbf{x}', -\mathbf{r}')$ to $(\mathbf{x}, -\mathbf{r})$. We can show this by considering each cases:

- Reflection: When reflection happens, $\|\mathbf{r}_\perp\|_2^2 \leq 2\Delta U$, and hence $\| - \mathbf{r}_\perp'\|_2^2 = \| - \mathbf{r}_\perp\|_2^2 = \|\mathbf{r}_\perp\|_2^2 \leq 2\Delta U$, meaning that the reverse trajectory would be reflected as well;
- Refraction: If $\Delta U > 0$, then the reverse trajectory is crossing the boundary from the other side, seeing a sudden decrease in potential, and hence would be refracted and regain the magnitude of momentum lost in the forward step. If $\Delta U < 0$, then $\| - \mathbf{r}_\perp'\|_2^2 > 2\Delta U$ and hence the reverse trajectory would also be refracted, ending in the original momentum with the sign reversed.

∎

Proposition 8, combined with time reversibility of leapfrog step as in basic HMC, shows that Voronoi sampling is time reversible. Hence we only need to use the Hamiltonian difference in the Metropolis-Hastings acceptance probability.

### E.2.4 Detailed Balance

**Theorem 1.** *A step of Structured Voronoi Sampling (Algorithm 4) satisfies the detailed balance.*

*Proof Sketch.* From measure preservation (App. E.2.1 and App. E.2.2), the change of variable as introduced by integrating Hamiltonian equations have Jacobian with absolute determinant 1 and hence can be omitted. From time reversibility (App. E.2.3), we only need to use the Hamiltonian difference in the Metropolis–Hastings acceptance probability. And, finally, by using a Metropolis–Hastings accepting step (lines 8 to 13) we ensure the detailed balance. ∎

### E.3 Related Work

Such an integrator scheme appears to be well-known in the computational physics community, dating back to as early as Jin and Wen [18], and quite possibly even earlier. Integration algorithms based on this idea continue to receive active research [44].

The first usage of such integrator with discrete stepin the context of MCMC appeared in Mohasel Afshar and Domke [30].[18] Mohasel Afshar and Domke [30] primarily based their motivation on the optical reflection–refraction analogy. We note that, in fact, Hamiltonian mechanics originated from Hamilton's formulation of geometrical optics (or Hamiltonian optics) [22, §VIII.7]. This connection, important to modern physics, should be interpreted with care in our context since momentum isn't a natural quantity in optical rays.

---

[18]In an earlier work, Pakman and Paninski [36] used the same idea but solved the trajectory exactly for simpler systems instead of using discrete steps.

Later, Nishimura et al. [33] provided a derivation of this integrator from the first principles of the calculus of variations [1]. Importantly, it should be noted that, due to the potential function being discontinuous, Eq. (16) loses its well-posedness as a system of differential equations, hence the necessity of justification based on non-smooth analysis.

## F   Algorithms

---

**Algorithm 2** Langevin Dynamics

---

**Input:** $\mathbf{x}^t$: current sample, $U$: potential energy function, $\varepsilon$: step size
**Output:** next sample: $\mathbf{x}^{t+1}$

1: $\mathbf{r} \sim \mathcal{N}(\mathbf{0}, \mathbb{I})$
2: $\mathbf{r} \leftarrow \mathbf{r} - \frac{\varepsilon}{2} \nabla U(\mathbf{x}^t)$
3: $\mathbf{x}^{t+1} \leftarrow \mathbf{x}^t + \varepsilon \mathbf{r}$
4: **return** $\mathbf{x}^{t+1}$

---

**Algorithm 3** MUCOLA

---

**Input:** $\mathbf{V}^t$: current sample, $U$: potential energy function, $\varepsilon$: step size
**Output:** next sample: $\mathbf{V}^{t+1}$

1: $\mathbf{r} \sim \mathcal{N}(\mathbf{0}, \mathbb{I})$
2: $\mathbf{r} \leftarrow \mathbf{r} - \frac{\varepsilon}{2} \nabla U(\mathbf{V}^t)$
3: $\mathbf{V}^{t+1} \leftarrow \mathrm{Proj}_{\mathcal{B}}(\mathbf{V}^t + \varepsilon \mathbf{r})$
4: **return** $\mathbf{V}^{t+1}$

---

**Algorithm 4** Structured Voronoi Sampling

---

**Input:** $\mathbf{x}^t$: current embeddings, $U$: potential function, $\varepsilon$: step size
**Output:** next sample: $\mathbf{x}^{t+1}$

1: $\mathbf{r} \sim \mathcal{N}(\mathbf{0}, \varepsilon \mathbb{I})$
2: $H^t \leftarrow U(\mathbf{V}_{\boldsymbol{m}^\star(\mathbf{x}^t)}) + K(\mathbf{r})$
3: $\mathbf{r} \leftarrow \mathbf{r} - \frac{\varepsilon}{2} \nabla U(\mathbf{V}_{\boldsymbol{m}^\star(\mathbf{x}^t)})$
4: $\mathbf{x}^{t+1} \leftarrow \mathrm{FINDDISC}(\mathbf{x}^t, \mathbf{r})$
5: $\mathbf{r} \leftarrow \mathbf{r} - \frac{\varepsilon}{2} \nabla U(\mathbf{V}_{\boldsymbol{m}^\star(\mathbf{x}^{t+1})})$
6: $H_{t+1} \leftarrow U(\mathbf{x}^{t+1}) + K(\mathbf{r})$
7: $\Delta H \leftarrow H^{t+1} - H^t$
8: **if** $\mathbf{s} \sim \mathcal{U}(0, 1) < e^{-\Delta H}$ :          ▷ *Metropolis criterion*
9:    **return** $\mathbf{x}^{t+1}$
10: **else**
11:    **return** $\mathbf{x}^{t+1} \leftarrow \mathbf{x}^t$

---

**Algorithm 5** REFRACTREFLECT

---

**Input:** $\mathbf{r}^t$: current momentum, $\boldsymbol{b}$ normal vector of the boundary, $\Delta U$: difference in potential energy
**Output:** next momentum: $\mathbf{r}^{t+1}$

1: $\mathbf{r}_\perp^t \leftarrow \frac{\boldsymbol{b}^T \mathbf{r}^t}{\|\boldsymbol{b}\|_2^2} \boldsymbol{b}$
2: $\mathbf{r}_\parallel^t \leftarrow \mathbf{r}^t - \mathbf{r}_\perp^t$
3: **if** $\|\mathbf{r}_\perp^t\|_2^2 > 2\Delta U$ :
4:    $\mathbf{r}_\perp^{t+1} \leftarrow \sqrt{\|\mathbf{r}_\perp^t\|_2^2 - 2\Delta U} \cdot \frac{\mathbf{r}_\perp^t}{\|\mathbf{r}_\perp^t\|_2^2}$          ▷ *refract*
5: **else**
6:    $\mathbf{r}_\perp^{t+1} \leftarrow -\mathbf{r}_\perp^t$          ▷ *reflect*
7: $\mathbf{r}^{t+1} \leftarrow \mathbf{r}_\perp^{t+1} + \mathbf{r}_\parallel^t$
8: **return** $\mathbf{r}^{t+1}$

---

| Split | Chinese | English | Fast food | French | Indian | Italian | Japanese | Total |
|-------|---------|---------|-----------|--------|--------|---------|----------|-------|
| train | 2929 | 4100 | 5891 | 5889 | 4412 | 5909 | 5996 | 42061 |
| valid | 1489 | 1780 | - | - | - | - | - | 4672 |
| test | 492 | 613 | 632 | 639 | 497 | 608 | 638 | 4693 |

Table 2: Number of restaurant reviews in each split and food type. Note that some reviews do not represent any specific food type, therefore, the total count is bigger than the sum count of reviews in each food category.

### F.1 Efficiently Finding Discontinuities

As explained in §6, the key insight in SVS is to adjust the momentum when facing discontinuity in the potential function. Therefore, we first need to find discontinuities efficiently and then reflect/refract on the boundary of the discontinuity. Remember that at each leapfrog step, we move from $\mathbf{x}^t$ to $\mathbf{x}^{t+1} = \mathbf{x}^t + \varepsilon \mathbf{r}$. To find discontinuities along this trajectory, we divide each leapfrog step into fractions of a step and check for discontinuity. Concretely, we only advance the embedding by a fraction $\alpha$, i.e. $\mathbf{x}' = \mathbf{x} + \alpha \varepsilon \mathbf{r}$ (line 4 in Algorithm 6). Then, we check for discontinuity by looking at the difference between the potential energies. If we don't observe any changes in the potential function, we take the next fraction of a step. Otherwise, we find the boundary and adjust the momentum (line 10 in Algorithm 6). Note that in SVS, finding the boundary is straightforward, and it is a hyperplane characterized by the normal vector, as the line goes through the two Voronoi centers on each side of the hyperplane.

---

**Algorithm 6** Find Discontinuity

---

**Input:** $\mathbf{x}^t$: current embeddings, $\mathbf{r}$: current momentum, $U$: potential function, $\varepsilon$: step size, $\alpha$: discontinuity step size
**Output:** next sample: $\mathbf{x}^{t+1}$

1: $\tau \leftarrow 0$
2: $\mathbf{x} \leftarrow \mathbf{x}^t$
3: **while** $\tau < 1$ :
4:     $\mathbf{x}' \leftarrow \mathbf{x} + \alpha \varepsilon \mathbf{r}$
5:     $\Delta U \leftarrow U(\mathbf{V}_{m^\star(\mathbf{x}')}) - U(\mathbf{V}_{m^\star(\mathbf{x})})$
6:     **if** $\Delta U = 0$ :                                                      ▷ *there is no discontinuity*
7:         $\mathbf{x} \leftarrow \mathbf{x}'$
8:         $\tau \leftarrow \tau + \alpha$
9:     **else**
10:        $\boldsymbol{b}, \alpha' \leftarrow \text{FINDB}(\mathbf{x}, \mathbf{x}')$      ▷ *Returns the intersection $\alpha'$ and the normal vector of the boundary $\boldsymbol{b}$*
11:        $\mathbf{x} \leftarrow \mathbf{x} + \alpha' \varepsilon \mathbf{r}$
12:        $\tau \leftarrow \tau + \alpha'$
13:        $\mathbf{r} \leftarrow \text{REFRACTREFLECT}(\mathbf{r}, \boldsymbol{b}, \Delta U)$
14: **return** $\mathbf{x}^{t+1} \leftarrow \mathbf{x}$

---

## G Dataset Statistics

This dataset is made available under the CC BY-SA 4.0 license. Our use of this dataset is for i) fine-tuning GPT-2, and ii) training classifiers for topic control, which clearly matches the intended use of this dataset mentioned by the creators as end-to-end training and generating text with content selection. A summary of dataset statistics can be found in Table 2.

## H Experimental Details

**Sampling algorithms.** Hyperparameters for each experiment are reported in Table 4. Following prior work [46], in algorithms based on Langevin dynamics, we apply an exponential decay to the step size by decreasing it to 0.05 after 500 steps. In all settings, we take 500 burn-in steps.

|  | Toy Example | | LM Sampling | | Controlled Generation | | |
|---|---|---|---|---|---|---|---|
|  | $\varepsilon$ | $\alpha$ | $\varepsilon$ | $\alpha$ | $\varepsilon$ | $\alpha$ | $\gamma$ |
| VS | 0.1 | 0.1 | - | - | - | - | - |
| SVS | 0.1 | 0.1 | 1. | 0.4 | 1.5 | 0.3 | 1.5 |
| HMC | 0.1 | 0.1 | - | - | - | - | - |
| LANGEVIN | - | - | 1. | - | 1.5 | - | 1.5 |
| MUCOLA | 0.1 | 0.1 | 1. | - | 1. | - | 2. |

Table 4: Hyperparameters used in sampling algorithms.

**Food Classifiers.** We train 3 classifiers. First, for experiments with FUDGE, we follow the experimental detail as in the original paper [48] and train a 3-layered BILSTM classifier with 0.5 dropout. The hidden dimension is set to 300, and the embedding layer is trained from scratch. Second, in experiments with MUCOLA, LANGEVIN, and SVS, to make the setup as close as to FUDGE, we train a 3-layered BILSTM classifier with 0.5 dropout. However, this time the BILSTM is trained on top of *frozen* GPT-2 representations, thus sharing the embedding layer that is necessary for these methods to work. Finally, to evaluate the success of the methods in following the control targets, we finetune a ROBERTA [25] base model. The accuracy of all the classifiers and the number of trained parameters are reported in Table 5. We train all the models on a single gtx_1080_ti GPU with approximately 2 hours of total computational budget.

Table 3: Inference time for different methods on the controlled generation experiments. All experiments are done on a single A100-40GB GPU.

| method | sec/batch |
|---|---|
| FUDGE | 10 |
| MUCOLA | 30 |
| LANGEVIN (ours) | 31 |
| SVS (ours) | 84 |

**Inference times.** We report inference times based on sec/batch. The number of decoded sentences per batch depends on the GPU memory and the size of the model. As depicted in table 3, using Voronoi measures does *not* increase the inference time (compare MUCOLA and LANGEVIN). We observe that SVS inference time is longer, because of the extra momentum adjustment steps. However, one can reduce the inference time by increasing $\alpha$.

| model | f1-score | precision | recall | # params |
|---|---|---|---|---|
| BILSTM | 0.87 | 0.87 | 0.87 | 17M |
| BILSTMPROBE | 0.84 | 0.84 | 0.84 | 37M |
| ROBERTA | 0.90 | 0.91 | 0.90 | 124M |
| BILSTMPROBE | 0.90 | 0.90 | 0.90 | 878M |

Table 5: Performance of food classifiers, and their number of learnable parameters, used in controlled generation experiment. All classifiers are trained and tested on E2E dataset.

# I  More Experiments on The Toy Model

To better understand the advantages of Voronoi sampling over HMC or MUCOLA, we further look at the JS divergence between the reference distribution and the distribution of samples when increasing the number of iterations. As depicted in Fig. 4, while the divergence decreases with more iterations across all sampling methods, Voronoi sampling converges to the reference distribution with fewer iterations. We then look at the distribution of sampled elements in Fig. 5. We observe that with 100 iterations, MUCOLA undersamples the element with the maximum probability while oversampling other elements. Finally, we extend the toy model from 4 squared cells in $\mathbb{R}^2$ to $2^k$ hypercube cells in $\mathbb{R}^k$ (Fig. 6). As the dimensionality increases, the divergence between the samples' distribution and the reference distribution also increases in all sampling methods. Importantly, Voronoi sampling consistently converges faster across different values for $k$.

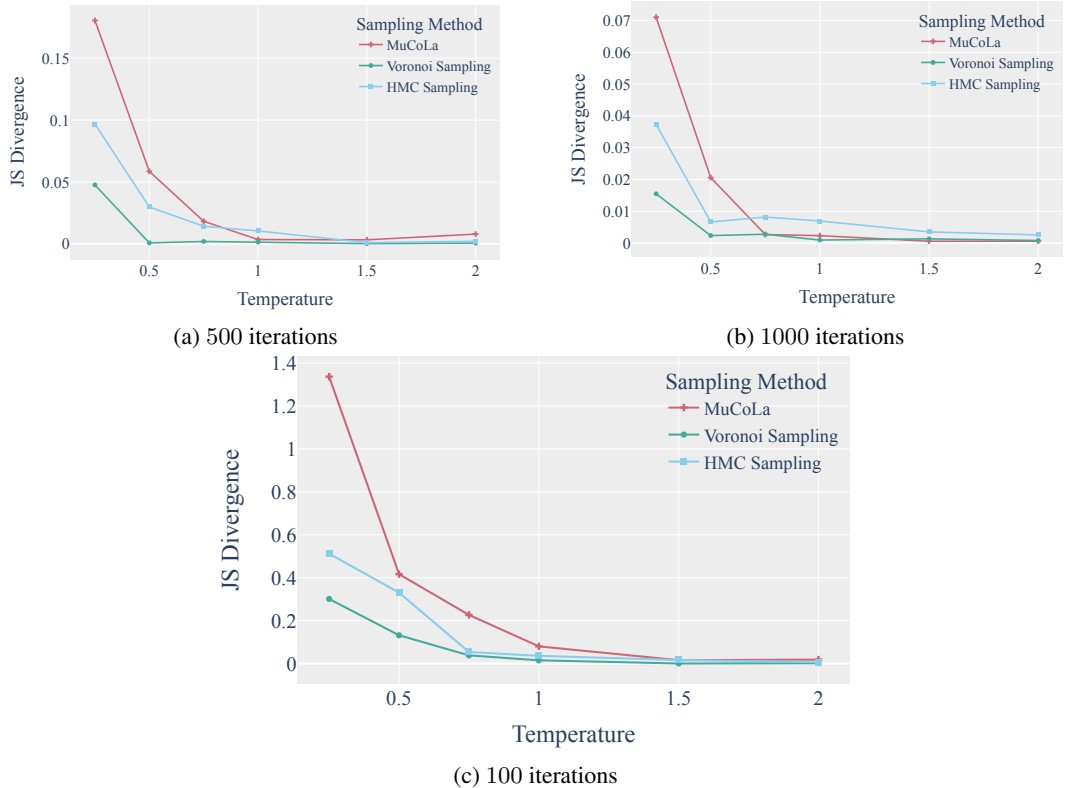

(a) 500 iterations

(b) 1000 iterations

(c) 100 iterations

Figure 4: Comparing JS divergence of methods using different numbers of iterations. In general, Voronoi sampling converge to the true distribution faster compared to HMC and MUCOLA. As the number of iterations increases, the divergence between the samples' distribution and the true distribution decreases across all sampling methods.

## J  Controlled Generation Results

Table 6, Fig. 7 show the performance of 4 sampling methods on different metrics. We show a sample of generations for each control target in Table 7.

| | FUDGE | | | | | MUCOLA | | | | | LANGEVIN | | | | | SVS | | | | |
|---|---|---|---|---|---|---|---|---|---|---|---|---|---|---|---|---|---|---|---|---|
| | Success | PPL | Dist-1 | Dist-2 | Dist-3 | Success | PPL | Dist-1 | Dist-2 | Dist-3 | Success | PPL | Dist-1 | Dist-2 | Dist-3 | Success | PPL | Dist-1 | Dist-2 | Dist-3 |
| Chinese | 0.30 | 5.41 | 0.38 | 0.53 | 0.63 | 0.30 | 10.90 | 0.21 | 0.35 | 0.47 | 1.00 | 16.21 | 0.22 | 0.36 | 0.48 | 0.90 | 12.42 | 0.22 | 0.36 | 0.49 |
| English | 0.15 | 5.82 | 0.41 | 0.57 | 0.67 | 0.45 | 17.76 | 0.26 | 0.39 | 0.49 | 0.70 | 15.82 | 0.27 | 0.43 | 0.55 | 0.85 | 15.00 | 0.25 | 0.41 | 0.54 |
| Fast food | 0.25 | 6.44 | 0.41 | 0.57 | 0.68 | 0.95 | 5.39 | 0.26 | 0.38 | 0.47 | 1.00 | 10.09 | 0.23 | 0.37 | 0.49 | 1.00 | 11.43 | 0.23 | 0.38 | 0.50 |
| French | 0.25 | 6.02 | 0.39 | 0.55 | 0.66 | 0.75 | 88.19 | 0.29 | 0.43 | 0.54 | 0.80 | 12.87 | 0.21 | 0.36 | 0.49 | 0.95 | 17.23 | 0.21 | 0.35 | 0.47 |
| Indian | 0.55 | 4.52 | 0.41 | 0.55 | 0.64 | 0.40 | 7.87 | 0.25 | 0.40 | 0.51 | 0.95 | 17.67 | 0.26 | 0.41 | 0.54 | 0.95 | 12.89 | 0.19 | 0.35 | 0.47 |
| Italian | 0.35 | 5.49 | 0.37 | 0.53 | 0.64 | 0.50 | 18.19 | 0.27 | 0.42 | 0.53 | 0.95 | 14.29 | 0.26 | 0.41 | 0.53 | 0.90 | 15.47 | 0.26 | 0.41 | 0.52 |
| Japanese | 0.30 | 5.44 | 0.42 | 0.55 | 0.65 | 0.75 | 83.37 | 0.27 | 0.42 | 0.54 | 1.00 | 12.90 | 0.21 | 0.36 | 0.47 | 0.95 | 12.89 | 0.18 | 0.33 | 0.45 |

Table 6: Evaluation of different sampling methods on controlled generation, using three criteria: success in following the control target (measured by the evaluator classifier), fluency (measured by perplexity), and diversity.

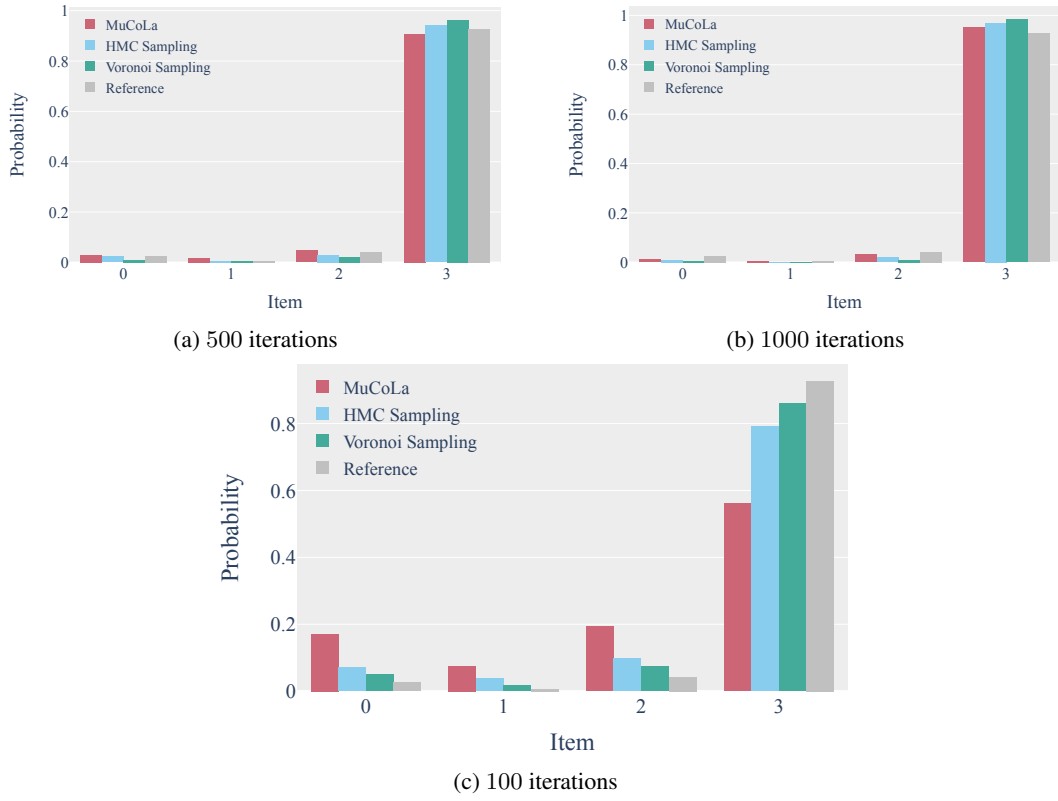

(a) 500 iterations

(b) 1000 iterations

(c) 100 iterations

Figure 5: Comparing the distribution of sampled elements at temperature $0.25$. With 100 iterations, MUCOLA undersamples the element with the highest probability while oversampling other elements.

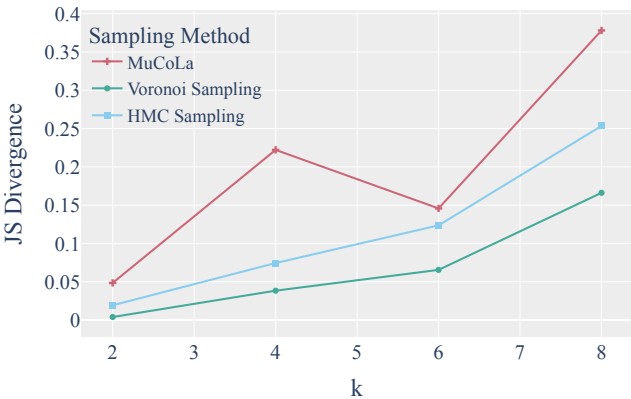

Figure 6: Comparing the distribution of sampled elements with the true distribution after 100 iterations, at temperature $0.5$. There are $2^k$ Voronoi cells with equal base measures in $\mathbb{R}^k$, where elements to sample from are located at the center of each Voronoi cell. Voronoi sampling converges faster to the true distribution across all $k$ values. As the dimensionality of the sample space increases, the divergence of all methods increases.

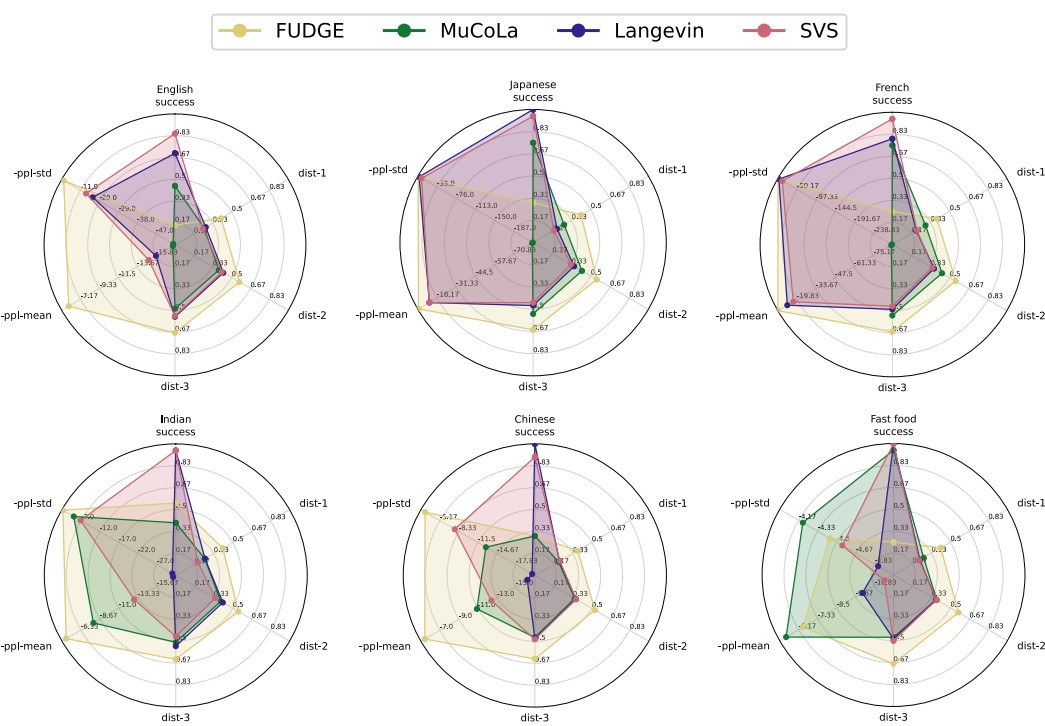

Figure 7: Evaluation of different sampling methods on restaurant review generation, along 6 axes: mean and standard deviation of negative perplexity[19](`-ppl-mean` ↑ , `-ppl-std` ↑), the percentage of generated sentences adhering to the control target (`success` ↑), and diversity metrics (`dist-1` ↑, `dist-2` ↑, `dist-3` ↑). For most control targets, SVS achieves the highest success rate, with relatively low perplexity.

| | **Chinese** |
|---|---|
| FUDGE | In the city centre near Yippee Noodle Bar Chinese, is Alimentum. It has moderate prices and |
| MuCoLa | and has a 1 out of 5. It has food and high customer rating. The Rice Boat is |
| LANGEVIN | It serves Chinese food with a low customer rating. The fast food and restaurant The Golden Curry is a |
| SVS | It has a low customer rating and a price. The highly rated Chinese restaurant The Phoenix has a high |

| | **English** |
|---|---|
| FUDGE | It has an average customer Rating. Bibimbap House has English food in the riverside area near |
| MuCoLa | and has a low customer rating. The Golden Curry is a children friendly, serving English food, with |
| LANGEVIN | It has low rating and is located near the to the city centre. The Phoenix is a English food |
| SVS | Alimentum in the city centre near the a moderate price range. It serves English food, is |

| | **Fast food** |
|---|---|
| FUDGE | A fast food, coffee shop, Strada has a low customer rating, has a price range of over £30. It is |
| MuCoLa | and is family friendly and serves fast food. The Wrestlers is a fast food coffee shop in the |
| LANGEVIN | It is located near the riverside, is a cheap family friendly fast food restaurant, and is called |
| SVS | It is located near the river. The Mill is a cheap, fast food and coffee shop near the |

| | **French** |
|---|---|
| FUDGE | It has a low-priced Inn French food. It is near Café Rouge.The Alimentum is a kid friendly fast food |
| MuCoLa | The French restaurant The Waterman is located in the city centre. The price range is less than |
| LANGEVIN | It is a restaurant located in the riverside, the restaurant, offers French food with a price |
| SVS | It is a family restaurant that serves French food with a price range and has a low customer rating. |

| | **Indian** |
|---|---|
| FUDGE | The Phoenix Indian restaurant has moderate prices with a 3 out of 5 rating. Located on the |
| MuCoLa | It is in the city and has a low customer rating. The Waterman is a low priced |
| LANGEVIN | It is not child friendly and it is near the river. It serves Indian food and a customer rating |
| SVS | It is located in the city centre near The Portland Arms Indian food and has a low customer rating. |

| | **Italian** |
|---|---|
| FUDGE | It has family Italian food and has a low a moderate price range. The Rice Boat has an average |
| MuCoLa | is a high priced Italian food restaurant with a customer rating of average. The Phoenix is a high |
| LANGEVIN | It is located in the city centre, it is not family friendly and is a coffee shop serving Italian |
| SVS | It is located in the the city centre near The Portland Arms.The Eagle is an Italian restaurant. |

| | **Japanese** |
|---|---|
| FUDGE | Japanese food. Its customer rating is 3 out of 5.The Phoenix is Japanese in the city centre |
| MuCoLa | for Japanese food is located in the city centre. It has a low customer rating. The Golden |
| LANGEVIN | It is located in the riverside. It is a Japanese food. It is a pub restaurant |
| SVS | It is located in the riverside. It is a low rated Japanese restaurant, and coffee shop. |

Table 7: Examples of sampled sentences from different control food targets.

