# OpenReview forum: "Structured Voronoi Sampling"
_NeurIPS.cc/2023/Conference — NeurIPS 2023 poster_

### Official Review · Reviewer_iTvg · 2023-06-23

**Soundness:** 3 good
**Presentation:** 2 fair
**Contribution:** 3 good
**Rating:** 5
**Confidence:** 2

**Summary:**

The paper pushes the frontier for gradient based sampling for auto regressive models. The paper lays clear theoretical issues to apply such techniques and address two issues: 1) the major contribution of the paper consists in proposing to construct voronoi like probability space over the output token embeddings to enable gradient based sampling process 2) the SVS algorithm also contains the novelty to deal with the non-continuity at the voronoi border by proposing a novel sampling algorithm.

The authors test the proposed algorithm at a toy setting, confirming their theoretical superiority transfers to this toy dataset over gradient sampling baselines. The authors also test their algorithm at generation and conditional generation tasks where the algorithm shows good performance in terms of success, perplexity and diversity.

**Strengths:**

The paper has made good contributions in making the gradient based sampling more practical to tackle real problems. The theoretical contribution of constructing voronoi distribution based on embeddings to perform HMS sampling is a significant one; furthermore, the paper addresses some practical issues via a novel sampling algorithm. The theoretical layout is also quite clear, well highlighting the solved issues.

The paper confirms their theoretical findings empirically as well as testing on two concrete NLP problems where the paper shows better than some existing NLP popular sampling techniques FUDGE.

**Weaknesses:**

My major problem for the paper is its presentation. The paper proposes a novel algorithm for sampling, however, all the algorithms have to be found in Appendix (minorly, these are not clearly indicated that it is in appendix for example for line 189). Note that the paper itself should be self contained and the appendix is really for interested readers to learn further details.

Another minor point for presentation is that the empirical part for this paper is relatively short and doesn't contain very detailed analysis. While the theoretical explanations is key for this paper, the similarity between eq(3) and eq(1) or eq(7) and eq(10) makes me think that the theoretical part can be shortened while maintaining the readability.

Finally, sampling is certainly not the only way (and arguably not the SOTA) way to perform conditional generation, RLHF used in InstructGPT or related techniques such as DPG (Khalifa et al. 2021) should be mentioned or in the best case compared to better situate the current work on its empirical aspect.

**Questions:**

In Table 1, all gradient sampling algorithms have a relatively high perplexity, which is reflected also in the examples in the appendix. Do the authors think it is an inherent limitation of this class of algorithms?

**Limitations:**

I appreciate the authors discuss the broader impact as well as transparently give insightful limitations.

The text quality issue is largely mitigated in modern large language models such as GPT-4 etc. This might inspire the authors might to rethink ways to improve on the quality aspect.

---

> ### Author Rebuttal · Authors · 2023-08-09
>
> Thank you for your feedback. We will make sure to move Algorithm 1. to the main body of the text. The reviewer suggested shortening the theoretical part to make space for experiments. We will make sure to add the additional experiments and contextualize them with the main text in the final manuscript. Since the main contribution is to offer a novel and principled way to apply gradient sampling for text generation, we devoted enough space to clearly explain our method. To this end, we need to systematically build up the knowledge that is not necessarily assumed to be known by all, which in fact is appreciated by the reviewers. We fear that shortening the theory part even further might confuse readers.
>
> > sampling is certainly not the only way (and arguably not the SOTA) way to perform conditional generation, RLHF used in InstructGPT or related techniques such as DPG (Khalifa et al. 2021) should be mentioned or in the best case compared to better situate the current work on its empirical aspect.
>
> In this work, we focus on vanilla language models. We agree with the reviewer that RLHF and instruction-finetuning change the probability distribution of the underlying LM, and can definitely impact the controllability of language models. Analyzing and adapting such models for controlled generation, however, needs further analysis which we leave as future work. We will discuss this point further in the Limitations section of the final manuscript.
>
> > In Table 1, all gradient sampling algorithms have a relatively high perplexity, which is reflected also in the examples in the appendix. Do the authors think it is an inherent limitation of this class of algorithms?
>
> Imposing a certain control on the generation process can introduce a trade-off between the success rate (in following the control) and fluency (that might be measured with perplexity). Such a trade-off exists not only for gradient-based sampling methods but also for other types of controlled generation methods (see Table 1. in [1]). Therefore, we do not believe that this is a particular limitation of gradient-based sampling methods.
>
> [1]  Alisa Liu, Maarten Sap, Ximing Lu, Swabha Swayamdipta, Chandra Bhagavatula, Noah A. Smith, and Yejin Choi. DExperts: Decoding-time controlled text generation with experts and anti-experts. In Proceedings of the 59th Annual Meeting of the Association for Computational Linguistics.

---

> > ### Comment · Reviewer_iTvg · 2023-08-15
> > **Thank you for the detailed answers**
> >
> > My comments have been addressed in the response, I hope that the presentation can be further improved when the paper is accepted to make the reading easier and more accessible by other readers.

---

> > > ### Author Response · Authors · 2023-08-18
> > >
> > > Thank you for taking the time to revisit the paper. We greatly appreciate it if you could also reconsider your score in light of the response and any new perspectives gained.

---

### Official Review · Reviewer_p6Uf · 2023-07-03

**Soundness:** 4 excellent
**Presentation:** 3 good
**Contribution:** 3 good
**Rating:** 7
**Confidence:** 4

**Summary:**

Authors have proposed a novel framework for gradient-based sampling from neural autoregressive LMs named Structured Voronoi Sampling. The core idea of the proposed approach is to map LM distribution to the embedding based version of it and then use newly proposed structured voronoi cells to perform sampling based on HMC. Authors carefully described each step along transformations and why it is theoretically sound. They discussed the conneciton of the proposed approach to related work such as COLD decoding and MuCoLa. They have done experiments in both toy task and controlled text generation to show the effectiveness and superiority of their approach.

**Strengths:**

# Originality

this work proposes a very coherent way of treating discrete neural based LMs using continuous densities. Opposed to prior work, their method uses less approximations and heuristics.

# Significance

Research problems around decoding strategies including sampling based decoding is essential given the wide spread of large scale language models. This work performs an important step towards better understanding of the gradient-based sampling methods and how to apply it to LMs parameterizing discrete/categorical distributions.

**Weaknesses:**

# Experiments

* Authors included samples from their methods and related work they have re-implemented, and these samples look pretty bad. Their proposed method repeats the same tokens right after each other making the sample look very unrealistic w.r.t. unknown data distribution e.g. "It is located in *the the* city centre near The Portland Arms.The Eagle is an Italian restaurant". This makes samples based results evaluations in the main text to be much less convincing. I tend to believe that the reason behind this is a poorly finetuned model: authors used GPT2 model which is quite out-dated while much better alternatives exist. Given that authors used modern GPUs (A100 40gb), they had all opportunities for choosing a stronger initial model.

* Authors used only 1000 samples to analyze empirical distributions which is quite limited considering the vocabulary size and avg sentence length. Using much more samples and a better trained model could reveal more interesting observations.

# Efficiency

The proposed approach is very slow even compared to Langevin dynamics. It would help to address this in the main text and not in the Appendix. Also it would help to put usual ancestral sampling time to see the gap between gradient based sampling and ancestral sampling in general.

**Questions:**

General comment: I believe this is very strong theoretical work here which has rather weak experimental setting because of poor choice of the initial model (GPT-2) for fine-tuning and the task setting. I think this could become a much stronger submission upon stronger experiments are done,

Questions:

sec 7.2 lines 293 - 299. Its unclear what do you mean by reference distribution here and how LM is related to it? IIUC you are saying that ancestral sampled from GPT-2 finetuned on E2E dataset resembles the unknown reference/data distribution? I think that is a very rough approximation. I think ancestral samples from GPT-2 gives unbiased estimates of sequence level distribution induced by GPT-2, thats it. The underlying connection between finetuned GPT-2 and reference distribution is unknown but likely not that strong given the samples you shown in the appendix.

Table 1: in the text you claim SVS outperforms everything else given metrics, but is it? It goes very close to Langevin per success and PPL (~ same given std), and diversity is lower than FUDGE / MuCoLa. Moreover, I think diversity should be compared to values computed over the data distribution, could you report that as well?

**Limitations:**

Authors included broader impact section and discussed how their framework could help to alleviate negative implications of large LMs as well as being a generator of intentionally toxic content.

---

> ### Author Rebuttal · Authors · 2023-08-09
>
> Thank you for your careful assessment and feedback.
>
> **Experiments**
>
> > Authors included samples from their methods and related work they have re-implemented, and these samples look pretty bad. Their proposed method repeats the same tokens right after each other making the sample look very unrealistic w.r.t. unknown data distribution e.g. "It is located in the the city centre near The Portland Arms.The Eagle is an Italian restaurant". This makes samples based results evaluations in the main text to be much less convincing. I tend to believe that the reason behind this is a poorly finetuned model: authors used GPT2 model which is quite out-dated while much better alternatives exist. Given that authors used modern GPUs (A100 40gb), they had all opportunities for choosing a stronger initial model.
>
> First, we should quickly clarify two points:
> - We must first note that we put a hard limit of only generating sequences of certain lengths, therefore, one must look at these generations as incomplete, the example that is mentioned in this review can take a complete form as: "It is located in the the city centre near The Portland Arms. The Eagle is an Italian restaurant with a customer rating of 1 out of 5."
> - Another important point to mention is the unavoidable trade-off between success and perplexity, which also impacts larger models.
>
> Besides the two points mentioned, we agree with the reviewer that using a larger model can help to have more fluent generations. To address this comment we added another experiment with GPT2-large, please refer to the response to all reviewers for further details.
>
> > Authors used only 1000 samples to analyze empirical distributions which is quite limited considering the vocabulary size and avg sentence length. Using much more samples and a better trained model could reveal more interesting observations.
>
> In general and in the absence of access to ground truth distribution, it is hard to know whether a certain number of samples is enough or not. We follow prior work, e.g., MuCoLa, in terms of choosing the number of generations.
>
> **Efficiency**
>
> We will consider moving the efficiency analysis to the main body. Comparing inference times to ancestral sampling, however, can be misleading since with ancestral sampling one can not enforce any constraints. In other words, there is a certain extra computation overhead that will be introduced (not only with gradient-based sampling but with any other method, e.g., FUDGE) if one needs to control an aspect in the generations.
>
> **Questions**
>
> > sec 7.2 lines 293 - 299. Its unclear what do you mean by reference distribution here and how LM is related to it? IIUC you are saying that ancestral sampled from GPT-2 finetuned on E2E dataset resembles the unknown reference/data distribution? I think that is a very rough approximation. I think ancestral samples from GPT-2 gives unbiased estimates of sequence level distribution induced by GPT-2, thats it. The underlying connection between finetuned GPT-2 and reference distribution is unknown but likely not that strong given the samples you shown in the appendix.
>
> The ground-truth distribution in that experiment is set to be the finetuned LM distribution. As the reviewer stated, ancestral samples give unbiased estimates of this distribution. Therefore, comparing the distribution of drawn samples with ancestral samples distribution can be helpful in understanding which sampling algorithms are less biased. Also, please note that the samples shown in the appendix are from the controlled generation setup and not ancestral samples of the fine-tuned model.
>
> > Table 1: in the text you claim SVS outperforms everything else given metrics, but is it? It goes very close to Langevin per success and PPL (~ same given std), and diversity is lower than FUDGE / MuCoLa. Moreover, I think diversity should be compared to values computed over the data distribution, could you report that as well?
>
> We do not claim that SVS significantly outperforms everything. Contributions of this paper result in two methods: Langevin (which operates on Voronoi measure) and SVS, and as mentioned in section 7.3 “both Langevin and SVS result in a high success rate and maintain fluency and diversity, and SVS is effective in maintaining a balance between various metrics and producing fluent sentences that adhere to control targets.” As we also highlight in that section, FUDGE gives more diverse samples but this comes at the cost of significantly lower success rates.
>
> Thank you for your suggestion on adding the ancestral samples results, we added that as the first row (GPT-2) in Table 8. in the additional uploaded PDF.

---

> > ### Comment · Reviewer_p6Uf · 2023-08-18
> > **thanks!**
> >
> > thanks for response! I am satisfied with the provided answers and extra experiments and going to increase my score!

---

### Official Review · Reviewer_qwXS · 2023-07-10

**Soundness:** 3 good
**Presentation:** 3 good
**Contribution:** 2 fair
**Rating:** 6
**Confidence:** 2

**Summary:**

The authors present Structured Voronoi Sampling, which is a gradient-based sampling approach. To be specific, the authors map the discrete distribution by a language model and defines densities; the density is then used to sample, which the process is based on hamiltonian monte carlo. The novelty of this paper comes from the theoretical analysis, but the core weakness is the empirical result; not much performance gain is seen across the experiments.

**Strengths:**

- The paper is well-written and structured. The flow of the paper is easy to follow, and the authors explain the introduced concept step-by-step.

- The paper proposes a novel gradient-based sampling method that caters for controlled generation tasks.

**Weaknesses:**

- Weak/wrong claims: i.e. Line 65-66. Not all language models share input and output embeddings. Authors should rephrase the sentence to avoid possible misunderstanding

- The empirical result is weak. There is not much of difference compared to Langevin in success and ppl score in Table 1. The same applies in Figure 3.


**Questions:**

Q1. Is there any particular reason why the authors did not test on popular controllable text generation task?

**Limitations:**

Please refer to summary and weakness section

---

> ### Author Rebuttal · Authors · 2023-08-09
>
> Thank you for your feedback.
> > i.e. Line 65-66. Not all language models share input and output embeddings. Authors should rephrase the sentence to avoid possible misunderstanding
>
> We will rephrase this as:  *_in **most** language models, the weights are shared between the language model head and the embedding layer_* in the final version of the manuscript.
>
> >The empirical result is weak. There is not much of difference compared to Langevin in success and ppl score in Table 1. The same applies in Figure 3.
>
> We need to clear a misunderstanding here: both Langevin and SVS methods in Table 1 operate on the Voronoi measure that is introduced in this paper, thus are contributions of this paper. Regarding the advantage of SVS over applying Langevin Dynamics directly on the Voronoi measure, we must note that it depends on the task and computational budget. Based on the insights gained from the Toy experiment, the difference is more pronounced when the underlying distribution is more peaky.
>
> > Is there any particular reason why the authors did not test on popular controllable text generation task?
>
> Unfortunately, there is no established and popular benchmark for controlled generation. We chose the same setup as used in [1]. To address your comment, we added a new experiment on a sentiment control task. Please refer to the general response to reviewers for more details.
>
> [1] ​​Xiang Lisa Li, John Thickstun, Ishaan Gulrajani, Percy Liang, and Tatsunori Hashimoto. Diffusion-LM improves controllable text generation. In NeurIPS 2022.

---

> > ### Comment · Reviewer_qwXS · 2023-08-18
> > **Response to Authors**
> >
> > The rebuttal reads well. There seems to be a misunderstanding by me in Langevin in the paper. Thank you for the rebuttal, and I will change my score accordingly.

---

### Official Review · Reviewer_BCGv · 2023-07-17

**Soundness:** 4 excellent
**Presentation:** 3 good
**Contribution:** 4 excellent
**Rating:** 8
**Confidence:** 4

**Summary:**

The paper proposes a new gradient-based sampling approach called Structured Voronoi Sampling (SVS) for controlled text generation. The key idea is to extend the discrete point distribution over word embeddings given by language models into a continuous density that spreads out probability over their corresponding Voronoi cells. A Hamiltonian Monte Carlo scheme is then devised to efficiently sample from that density, handling discontinuities between cells through a volume-preserving refraction/reflection trick. Empirical results on a toy problem and a more realistic controlled text generation problem show SVS can better match target distributions and control constraints compared to baselines.


**Strengths:**

Developing provably sound sampling methods for text generation remains an important open problem for language models, especially for controlled generation. The proposed  method is well-motivated from first principles and provides formal guarantees unavailable with prior heuristic approaches, with rigorous mathematical derivations provided in appendices.

A number of innovative steps are taken to get the core Voronoi idea to work, including (i) lifting the discrete token distribution to the embedding space, (ii) smoothing the measure-zero discrete distribution to a continuous density with the Voronoi transformation, and (iii) handling discontinuities in sampling with the refraction/reflection trick. These moves are independently interesting in their own right and a large segment of the NeurIPS community will likely find at least one of them to be novel and potentially applicable in future work.

Related methods like MuCoLa and Fudge are discussed and compared in the experiments. The toy domain is pedagogically useful for exposing the limitations of existing approaches, while the scaled-up controlled generation results are promising, showing competitive fidelity and control.

The paper is well-written and structured overall, and the writing systematically builds up the relevant concepts for a typical NeurIPS reader to understand the significance. The key generalizable ideas are appropriately emphasized.

**Weaknesses:**

(1) The most significant issue limiting the applicability of SVS as currently posed is the problem of calculating the base measure $\mu$ in a high-dimensional space. This problem in some sense puts us back where we started, as the original problem motivating SVS in the first place is that the integral in the normalizing constant of the energy function is not tractable. Computing the exact integral for the base measure in SVS is obviously also not tractable (as noted up-front as a limitation the paper, which I appreciated). The assumption in 6.1 that all cells have equal base measure is a very strong assumption, given that this clearly does not hold in practice, and somewhat undermines the very careful and rigorous progression of proofs of correctness.

I see a couple way to strengthen this part of the paper, and would strongly recommend doing something to close this gap for the paper to be maximally impactful. One possibility is to provide some analysis of the consequences of violating this ‘equal base measures’ assumption: is there a bound on how bad violations can get? how badly is this assumption violated in the controlled generation task being reported? Intuitively, ignoring real diff them would result in over- or under-representing certain regions. A second possibility would be to take a first step toward some approximate method to account for base measure differences. For example, just taking distances to the first k nearest neighbors of a point could efficiently put a bound on how big the cell could be? Or some kind of Monte Carlo approximation? Clearly a full resolution of this problem is a task for another paper, and I do not expect this problem to be fully solved, but charting a course (even if that course is a bit inefficient) would go a long way.

(2) More analysis could be provided on the toy example with known reference distribution; for example, what do the sampled distributions actually look like? In what way is MuCoLa off when temperature is low? How does this scale with the size of the space (e.g. if instead of a 2x2 square, it were a 2^k hypercube in k dimensions)? Does MuCoLa just take longer to burn in? or gets stuck sampling {0,1,2}? I would have liked to understand these failure modes a bit better to better motivate how the proposed method overcomes them.

(3) The controlled generation task could also be strengthened: the results are from a single fairly bespoke 'food' task and classifier, which limits generalizability. It would be more compelling to evaluate on a more systematic suite of controlled language generation benchmarks to better pinpoint where the benefits of Voronoi sampling are most pronounced. Additionally, since controlled generation is highly dependent on the (trained) classifier being used to guide sampling, it would be helpful to test lower or higher capacity classifiers with different uncertainty levels. I worry that using the same classifier for both generation and evaluation (the ‘success’ column of Table 1) means we may just be measuring the sampler’s ability to overfit to a bad classifier (it is quite a bit more ‘faithful’ to this classifier than MuCoLa or Fudge, which is not necessarily a good thing if it is just ‘hacking’ a bad classifier).

**Questions:**

1. More exposition could be provided to unpack eq. (2) which may be confusing to a typical neurips reader. in particular, I think many practitioners are more familiar writing the final layer as a softmax applied to a linear transformation, i.e. something like $P(w_n | w_{<n}) = softmax(v * f(w_{<n}))$ where $v$ is the full embedding matrix). Some readers may also have forgotten that GPT2 used the same embedding matrix at the initial and final layers (other implementation use a separate fully-connected layer here), so it may be worth a gentle reminder to avoid confusion.

2. Similarly, some additional clarification could be given to interpret the denominator in Def. 2 and 3 as is essentially normalizing by the ‘volume’ of the cell (with the measure giving a suitable notion of volume). This will be obvious to those versed in measure theory but could help other ML researchers with less mathematical background.

3. I was trying to think through the precise relationship to other approaches based on nearest-neighbor smoothing (e.g. Khandelwal [et](http://U.et) al, 2019. “*Generalization through memorization: Nearest neighbor language models*.” or Khandelwal et al, 2020. “*Nearest neighbor machine translation”;* El-Kishky et al, 2023. “*kNN-Embed: Locally Smoothed Embedding Mixtures for Multi-interest Candidate Retrieval”*). These approaches have a very similar flavor to the voronoi embedding, which on the surface appears to be equivalent to some kind of importance sampling based on nearest-neighbors. Does the voronoi sampler reduce to something like this as a special case?

4. very minor: It was a bit confusing to have Alg. 1 prominently referred to in the main text (down to giving line numbers) but not be able to find it until I was later going through the supplemental. if it is important, it may be worth trying to fit into the main text.

5. very minor: the abstract uses the acronym SVS but it doesn’t reappear until Fig. 3, at which point I was very confused (pragmatically, it’s confusing that it’s referred to as ‘voronoi sampling’ in Fig. 2, right next to it, which implies that SVS is something different?)

6. very minor: is it worth pointing out in section 7.2 that there is no reason to actually prefer voronoi over regular ancestral sampling, given that ancestral sampling is already very efficient and provably matches the target distribution? And that we’re just doing this exercise (Fig. 3) as validation that the voronoi sampler is pretty close to the ‘gold standard’?

**Limitations:**

See above.

---

> ### Author Rebuttal · Authors · 2023-08-09
>
> Thank you for the careful analysis of our work and your valuable suggestions.
> (1) Regarding the assumption on base measures and approximations:
> thank you for your suggestions on how to strengthen the paper. Perhaps the way that is more practical to do it to approximate base measures is through importance sampling.
>
> An integration of importance sampling with SVS can look like this: at each reflection/refraction step (line 3 in Alg. 5), we need to compute the difference in the potential energy of two points:
> - If these points belong to the same Voronoi cell, then the base measures are equal.
> - If not, let’s suppose they belong to cells $C_m$ and $C_{m’}$. We take a number of samples from a Gaussian distribution with the mean set to the center of the corresponding Voronoi cells (let's call them $m$ and $m'$).
> We then approximate $\int_{C_m} \mathrm{exp}(-\frac{1}{2}  \| g_m^t - x \|^2) = \int_{C_m} f(x) $ with $\frac{1}{N} \sum_{n=1}^N \frac{f(x)}{q(x)}$ where $q \sim \mathcal{N}(m, \varepsilon I)$. Ideally, one needs to use truncated Gaussian distribution as $q$, however, we might use the Gaussian distribution as an approximation.
>
>
> As the reviewer stated, analyzing and improving this approximation can be the subject of a separate project that we defer to future work.
> (2) Regarding extending the analysis on the toy experiment: thank you for this suggestion. We added more experiments on the toy model. Here is a summary of what we found:
> - In Figure 5, we vary the number of iterations. We observe that the difference between sampling methods is more pronounced with fewer iterations.
> - As suggested, we then look into the distribution of sampled elements at temperature 0.25 after 100 iterations. We observe that MuCoLa is sampling the highest probability element less frequently than Voronoi sampling or the reference distribution, while sampling {0, 1, 2} more often than needed.
> - Finally, we extend the toy model to hypercubes in k dimensions (figure 7). Generally speaking, as the dimensionality increases, the divergence between the samples’ distribution and the true distribution also increases in all sampling methods. Furthermore, Voronoi sampling consistently converges faster across different values for k.
>
> (3) Regarding the choice of the controlled generation task and classifiers: we added another experiment on a common task, please refer to the general answer to all reviewers for further details.
>
> Indeed, the classifier’s accuracy and certainty have a significant role in the generation quality. To ensure fairness as much as possible, we use the exact same classifier for the gradient-based algorithms, and a similar architecture for FUDGE.
>
> We also need to clear a misunderstanding here: the success measures reported in Table 1, are **not** evaluated by the same classifier used for controlling the generations. We train a separate and arguably more accurate “evaluator” classifier. Please refer to Table 5 for comparing the accuracies of classifiers.
>
> **Questions**
>
> Thank you for your suggestions, we will add the suggested clarifications and move Algorithm 1. to the main body of text in the final version of the manuscript.
>
> > I was trying to think through the precise relationship to other approaches based on nearest-neighbor smoothing (e.g. Khandelwal et al, 2019. “Generalization through memorization: Nearest neighbor language models.” or Khandelwal et al, 2020. “Nearest neighbor machine translation”; El-Kishky et al, 2023. “kNN-Embed: Locally Smoothed Embedding Mixtures for Multi-interest Candidate Retrieval”). These approaches have a very similar flavor to the voronoi embedding, which on the surface appears to be equivalent to some kind of importance sampling based on nearest-neighbors. Does the voronoi sampler reduce to something like this as a special case?
>
> We agree with the reviewer that on the surface these approaches might seem similar, however, there are major differences between Voronoi sampling and kNN-LMs that makes us believe that none is a special case of another. First, in kNN-LMs, a training dataset (or a set of examples) is cached, which will be then used during the generation. However, in this work, the centers of Voronoi cells are simply the words in the vocabulary. Second, sampling a new word in kNN-LM is still autoregressive and the underlying probability distribution is an interpolation between the LM probability and the distance to the cached exemplars. However, in Voronoi sampling, we sample the whole sequence at once, and the probability of sampling a sequence is an unaltered LM probability.
>
> > very minor: the abstract uses the acronym SVS but it doesn’t reappear until Fig. 3, at which point I was very confused (pragmatically, it’s confusing that it’s referred to as ‘voronoi sampling’ in Fig. 2, right next to it, which implies that SVS is something different?)
>
> We intentionally use different terms in these two figures. In the toy experiment, only one embedding is sampled, which is different from the text generation experiments where a sequence of embeddings is sampled. Therefore, we call the former Voronoi sampling and the latter structured Voronoi sampling to highlight this difference. We will clarify this further in the final version of the manuscript.

---

> > ### Comment · Reviewer_BCGv · 2023-08-15
> > **Thanks!**
> >
> > I appreciate the thoughtful response and the improvements described, which strengthen an already very-strong paper.

---

### Official Review · Reviewer_v2AJ · 2023-07-25

**Soundness:** 4 excellent
**Presentation:** 3 good
**Contribution:** 3 good
**Rating:** 6
**Confidence:** 3

**Summary:**

This paper proposes Structured Voronoi Sampling: a new gradient-based algorithm to sample from a distribution (i.e. a language model). The proposed approach leads to comparably fluent text whilst being able to better follow constraints (e.g. a topic) for the desired generation.

**Strengths:**

1. a new sampling method that is a correct MCMC scheme (unlike previous gradient-based samplers)
2. evaluation on a synthetic task shows that the proposed approach better models the tail of the true distribution
3. evaluation on constrained language modeling shows that the proposed approach achieves good fluency and diversity, whist being able to follow the control target

**Weaknesses:**

1. while the writing is generally great, it would help to have an introduction of gradient-based sampling before Section 3
2. it is not clear how the proposed approximation of the (costly!) base measure affects the correctness of the proposed method

**Questions:**

1. some of the key algorithms and results are reported in the appendix. It would be good to include them in the main text upon acceptance
2. in eq (2), enc(context) is a single vector. This is not the case with current models (e.g. GPT-2) which have a vector per input token. Can you comment on how this affects the formulation of LM with Embeddings?
3. in eq (3), the conditioning on V_{<n} seems wrong, as `n` is only defined in the Cartesian product

**Limitations:**

yes.

---

> ### Author Rebuttal · Authors · 2023-08-09
>
> Thank you for your feedback.
>
> Regarding the suggestion on the presentation of gradient-based sampling, we will try to motivate gradient-based sampling further before going to the details in the final manuscript.
> > it is not clear how the proposed approximation of the (costly!) base measure affects the correctness of the proposed method
>
> We do not apply any approximation in our experiments. As stated in the paper, our approach offers an exact sampling algorithm, up to the computation of the difference in base measures. This paves the way for future works on potential strategies to approximate the difference in base measures. Our empirical findings indicate that SVS exhibits a strong performance when compared to MuCoLa.
>
> **Questions**
> 1. We will move Algorithm 1 to the main body in the final manuscript.
> 2. $\mathrm{enc}(w_{<n})$ is a single vector and can be the output of current LMs (like GPT-2). Concretely, to compute this vector with GPT-2, we pass the context $w_{<n}$ to GPT-2, and look at the last layer representation of GPT-2 **at position $n$**. We will make sure to clarify this further in the final version of the manuscript.
> 3. Thank you for catching the typo, the conditioning should be on V. We will fix this in the final manuscript.

---

> > ### Comment · Reviewer_v2AJ · 2023-08-20
> > **Feedback on rebuttal**
> >
> > Thank you for clarifying my concerns and engaging with the comments raised by the other reviewers!

---

### Official Review · Reviewer_tTqL · 2023-07-27

**Soundness:** 3 good
**Presentation:** 3 good
**Contribution:** 3 good
**Rating:** 7
**Confidence:** 3

**Summary:**

Gradient-based sampling for text generation is an important challenge, as it allows for sampling from energy-based models, such as one defined by a mixture of experts as found in classifier-guided sampling. The main challenges in gradient-based sampling for discrete distributions are encoding the discrete distribution into $R^d$ and dealing with inevitable discontinuities that arise in the encoding. The paper proposes a method that addresses these two challenges with Voronoi measures and an application of refract+reflect HMC respectively.

The method is validated in three settings: First, sampling from a non-structured and tractable discrete distribution with 4 classes and associated embeddings. Second, sampling from a language model. Third, sampling from a language model with additional constraints. In all settings, the method improves upon baselines, namely Fudge, MuCoLa, and Langevin dynamics (without reflection).

**Strengths:**

* The approach is well-motivated and interesting.
* The writing is clear and easy to follow.
* In text generation, unconstrained and constrained, the method does show improvements over the MuCoLa baseline. However, the gains from reflection and refraction seem quite small.

**Weaknesses:**

Experiment baselines: The paper only compares to MuCoLa and Fudge as text generation baselines, but other methods for controllable text-generation exist such as diffusion-LMs and other Gibbs with gradient (GwG) methods [2]. Since the claim is principled gradient-based methods for text-generation, not having a comparison to diffusion-LMs seems reasonable. However, I believe other GwG methods should be compared (see questions).

**Questions:**

1. How is the method related to Gibbs with gradients methods, such as [1] and [2]? SVS takes advantage of embedding geometry and I believe GwG does not, which would plausibly lead to improvements.
2. It would have been nice to see comparisons to non-principled gradient-based sampling, such as COLD. A lack of rigorous justification for an existing method should not be enough to discount its (potential) effectiveness.
3. Why is the constrained text generation task different from the tasks studied in MuCoLa?
4. There are quite a few references to lines of algorithms in the Appendix.
5. Are the small gains from reflection due to the length of the chains? Would the difference between Langevin and SVS be larger if the computational budget was smaller?

[1] Grathwohl, W., Swersky, K., Hashemi, M., Duvenaud, D.K., & Maddison, C.J. Oops I Took A Gradient: Scalable Sampling for Discrete Distributions. ICML 2021.

[2] Zhang, R., Liu, X., & Liu, Q. (2022). A Langevin-like Sampler for Discrete Distributions. ICML 2022.

**Limitations:**

The limitations were adequately addressed.

---

> ### Author Rebuttal · Authors · 2023-08-09
>
> Thank you for bringing up the Gibbs with gradient methods. We will ensure their discussion in the related works section of the final manuscript. However, in terms of empirical evidence, [1] doesn't feature any experiments on language generation. We believe that applying this approach to sample a text sequence would be impractical without further improvements. The primary reason is that both [1] and [2] operate within the logits domain. In the context of language generation, this entails sampling each element from a vocabulary of approximately 50,000 items (as in the case of GPT-2 small). This vocabulary size is significantly larger compared to MuCoLa or SVS, which function in the embedding space $\mathbb{R}^{768}$. While [2] does present an experiment on the infilling task, it only samples 25% of tokens within sentences. This setup is computationally less demanding than the generation settings outlined in our paper. To the best of our knowledge, the code for the infilling experiment hasn't been released. Consequently, additional research and implementation are necessary to scale these methods for sampling sequences of length 20 or more.
>
> **Questions**
> 1. The benefit of Gibbs with gradient methods is that they provide bounds for convergence. However, as stated by the reviewer, they do not benefit from the similarity of the word embeddings when navigating the state space. As mentioned above, such approaches are considerably more expensive, which limits their applicability in real-world text generation settings.
> 2. We’ve attempted to adapt COLD for the controlled generation experiment. Unfortunately and even after increasing the control term’s weight, the outcomes closely resembled those of the uncontrolled GPT-2, resulting in very low success rates. We plan to persist in our efforts to improve COLD results by experimenting with varying hyperparameters. If we observe any improvement, we will include the updated results in the final manuscript.
> 3. Unfortunately, there is no widely used benchmark for controlled generation. Even when two papers use the same benchmark, the exact prompts that are used could be different, which limits the reproducibility and fairness of the results. In this paper, we followed the experimental setup in [3]. To address your comment, we added a new experiment on a task that is studied in MuCoLa and other prior works, please refer to the general response to the reviewers for further details.
> 4. We will move Algorithm 1. to the main body of text in the final manuscript.
> 5. Yes, that could very well be the case. However, it is hard to empirically test this hypothesis on the language generation experiments, since the ground truth distribution is unknown. To answer this, we added more experiments on the toy model (please see Figure 5) that supports this hypothesis. We observe that SVS has the lowest JS divergence compared to HMC on the Voronoi measure, and the difference is more pronounced when doing fewer iterations.
>
> [3] ​​Xiang Lisa Li, John Thickstun, Ishaan Gulrajani, Percy Liang, and Tatsunori Hashimoto. Diffusion-LM improves controllable text generation. In NeurIPS 2022.

---

> > ### Comment · Reviewer_tTqL · 2023-08-17
> >
> > My initial score did not reflect the strength of the paper, and will be increased to accept.
> >
> > Regarding comparisons to Gibbs with gradient (GwG) methods: More experiments would be very nice to have, but not necessary. The proposed method is more general than sampling from only an embedding-parameterized language model, and therefore would ideally also be compared to GwG methods in at least a toy setting. The likeliness of the proposed method outperforming GwG (due to the additional assumption of access to embeddings) is an opportunity to broaden the impact and strengthen the paper, rather than a weakness. One possible experiment would be to show error and runtime / steps for an embedding-parameterized model from [1] or [2], such as a Potts model, at various numbers of classes / embedding dimensions.
> >
> > [1] Grathwohl, W., Swersky, K., Hashemi, M., Duvenaud, D.K., & Maddison, C.J. Oops I Took A Gradient: Scalable Sampling for Discrete Distributions. ICML 2021.
> >
> > [2] Zhang, R., Liu, X., & Liu, Q. (2022). A Langevin-like Sampler for Discrete Distributions. ICML 2022.

---

> > > ### Author Response · Authors · 2023-08-18
> > >
> > > Thank you for reading our response, and your great suggestion on comparing SVS and GwG on a toy model. We will work on this and consider adding it to the final version of the manuscript.

---

### Author Rebuttal · Authors · 2023-08-09

We thank the reviewers for providing valuable and constructive feedback. We first provide responses to a shared concern raised by multiple reviewers. Responses to individual reviewers are provided below. We report the results of new experiments in an additional PDF.

Multiple reviewers were concerned that we could have picked a more popular task for the controlled generation experiment. To address this concern, we added a new experiment on a more popular sentiment control task, which many of the prior works also experimented with [1, 2, 3]. The goal of the task is to control the sentiment of the generations.

We use the same 15 prompts used in [1, 2] and generate 10 samples per prompt using **GPT2-Large**. Similar to prior work, we train classifiers on SST-2 dataset for sentiment classification. We use this classifier to enforce a positive sentiment in the generations. Results are shown in Table 8. of the additional PDF. We observe the following:
- FUDGE mostly fails in following the control, while providing more fluent and diverse outputs.
- MuCoLa achieves a higher success rate compared to FUDGE, but significantly lower success rates compared to Langevin or SVS, with a high variance in perplexity and success rate.
- Both SVS and Langevin Dynamics on the Voronoi measure perform quite well in terms of following the control, and SVS achieves the best overall success rate.

These results are more or less in line with the topic control experiment, i.e., Table 1.


[1] Plug and play language models: A simple approach to controlled text generation. In International Conference on Learning Representations (ICLR), 2019.

[2] Sachin Kumar, Biswajit Paria, and Yulia Tsvetkov. Constrained sampling from language models via Langevin dynamics in embedding spaces. In Proceedings of the 2022 Conference on Empirical Methods in Natural Language Processing (EMNLP).

[3] Alisa Liu, Maarten Sap, Ximing Lu, Swabha Swayamdipta, Chandra Bhagavatula, Noah A. Smith, and Yejin Choi. DExperts: Decoding-time controlled text generation with experts and anti-experts. In Proceedings of the 59th Annual Meeting of the Association for Computational Linguistics.

---

### Decision · Program_Chairs · 2023-09-21

**Decision:**

Accept (poster)

**Comment:**

Throughout the review, the reviewers acknowledge the innovative method using Voronoi measure and reflection-refraction treatment to allow a gradient-based MCMC for sampling discrete variables, and the empirical benefits on various discrete conditional generation tasks. The reviewers are also concerned about comparison with recent alternative methods, the assumption of equal base measure over cells, and the application to more interested language models, for which the authors seem to have satisfyingly addressed in the rebuttal.

Nevertheless, I found a theoretical inconsistency in the methodology part. In implementing the method in Section 4, the proposed base measure in Eq. (12) requires the gradient $g_m := \nabla_V \log p(V)$. But the definition of this gradient is confusing: according to the words under Eq. (3), "$p(V)$ only places a positive probability on a countable set", and in Section 2.2., "importantly and similar to $p(V)$, $p(V \mid t)$ only places a positive probability on a countable set", hence $p(V)$ and $p(V \mid t)$ are not absolutely continuous w.r.t the Lebesgue measure and does not even have a valid density function. Such an explanation is also mentioned in Section 5.2 as the limitation of the MuCoLa method. If the gradient is to be understood as the gradient w.r.t each $v_w$ of Eq. (2), then the contribution is made less clear since one can directly apply LD and HMC using this $\nabla_V \log p(V)$.
Moreover, I do not think it is "natural" to target $x$ __towards__ $g_m$, but to let $x$ __follow__ $g_m$. In other words, I think it is more natural to take the gradient as $g_m$ in Eq. (14) instead of $g_m - x$.
A further minor issue is that, as the central idea of the method is to extend the measure on a finite (or at least countable) set of points to the continuous space, comparison to alternative well-established approaches to do this is expected, e.g., the Gumbel-Softmax trick [https://openreview.net/pdf?id=rkE3y85ee].

I posted these concerns to all the reviewers and called for discussions. I got feedback that $g_m - x$ in Eq. (14) is indeed unnatural from one reviewer. But two reviewers expressed the opinion that the innovation and empirical results could outweigh the technical flaws, which can be fixed by revising the writing. I hence give an accept following these opinions. But the authors must revise the paper to clearly address the two technical problems, by e.g., redefining $g_m$ by introducing a distribution variant that spreads density over the entire embedding space, and use $g_m$ instead of $g_m - x$ in Eq. (14).